EMBO
Molecular Medicine

# Thioredoxin-interacting protein regulates protein disulfide isomerases and endoplasmic reticulum stress

Samuel Lee[1,2,3], Soo Min Kim[1,2], James Dotimas[1,2], Letitia Li[1,2], Edward P Feener[4], Stephan Baldus[3], Ronald B Myers[1,2], William A Chutkow[2], Parth Patwari[2], Jun Yoshioka[2] & Richard T Lee[1,2,*]

## Abstract

The endoplasmic reticulum (ER) is responsible for protein folding, modification, and trafficking. Accumulation of unfolded or mis-folded proteins represents the condition of ER stress and triggers the unfolded protein response (UPR), a key mechanism linking supply of excess nutrients to insulin resistance and type 2 diabetes in obesity. The ER harbors proteins that participate in protein folding including protein disulfide isomerases (PDIs). Changes in PDI activity are associated with protein misfolding and ER stress. Here, we show that thioredoxin-interacting protein (Txnip), a member of the arrestin protein superfamily and one of the most strongly induced proteins in diabetic patients, regulates PDI activity and UPR signaling. We found that Txnip binds to PDIs and increases their enzymatic activity. Genetic deletion of Txnip in cells and mice led to increased protein ubiquitination and splicing of the UPR regulated transcription factor X-box-binding protein 1 (Xbp1s) at baseline as well as under ER stress. Our results reveal Txnip as a novel direct regulator of PDI activity and a feedback mechanism of UPR signaling to decrease ER stress.

**Keywords** endoplasmic reticulum stress; protein disulfide isomerases; thioredoxin-interacting protein; unfolded protein response
**Subject Category** Metabolism

## Introduction

The endoplasmic reticulum (ER) is an organelle that participates in metabolic pathophysiologies including insulin resistance and type 2 diabetes (Hotamisligil, 2010). The ER is responsible for folding, modification, and trafficking of a large number of secreted and membrane proteins, and it is part of a very dynamic system capable of quickly adapting to changes in metabolic and synthetic cellular demand. To maintain its functional integrity, the ER has to constantly balance the capacity of its protein chaperones with the load of unfolded proteins that are newly synthesized by the cell. A mismatch between the load of unfolded proteins and the capacity of protein chaperones in the ER results in the accumulation of unfolded or misfolded proteins in the ER; this condition has been termed 'ER stress' (Hotamisligil, 2010). Upon ER stress, three different response pathways are activated that are collectively called the 'unfolded protein response' (UPR). This short-term adaptive system is necessary to maintain balance in the protein folding machinery by activating protein modification and gene expression programs designed to inhibit protein translation, increase the production of protein chaperones, and stimulate protein degradation. However, chronic UPR activation by ER stress can be detrimental and may lead to maladaptive changes in cellular signaling that ultimately contribute to disease processes including insulin resistance and type 2 diabetes (Hotamisligil, 2010; Samuel & Shulman, 2012).

Protein disulfide isomerases (PDIs) are oxidoreductases that are responsible for the introduction of disulfide bonds into proteins through thiol-disulfide exchange reactions. A common feature of most PDIs is that they contain at least one thioredoxin-like fold with a CXXC redox active motif (Hatahet & Ruddock, 2009). A study that investigated the consequences of S-nitrosylation on PDI function showed that changes in PDI activity lead to increased protein misfolding and ER stress and contribute to neurodegeneration, a central feature of Parkinson's and Alzheimer's disease (Uehara et al, 2006). This provides proof of principle that regulation of PDI activity is relevant for ER stress-related disease progression.

Thioredoxin-interacting protein (Txnip) is a protein that binds directly to thioredoxin and regulates redox signaling in the cell (Nishiyama et al, 1999). Accumulating evidence demonstrates that Txnip, like other members of a protein family sometimes called the 'alpha arrestins', serves as a multifunctional adaptor protein for different signaling pathways (Lee et al, 2013). In addition to regulating

1  Harvard Department of Stem Cell and Regenerative Biology, Harvard Stem Cell Institute, Harvard Medical School, Brigham and Women's Hospital, Cambridge, MA, USA
2  The Cardiovascular Division, Department of Medicine, Harvard Medical School, Brigham and Women's Hospital, Cambridge, MA, USA
3  Department III of Internal Medicine, University Hospital of Cologne, Cologne, Germany
4  The Joslin Diabetes Center, Harvard Medical School, Boston, MA, USA
   *Corresponding author. Tel: +1 617 768 8282; Fax: +1 617 768 8280; E-mail: RLee@partners.org

cellular redox state, Txnip also regulates metabolism in diverse cell types; we and others have shown that Txnip deficiency decreases gluconeogenesis and increases lipogenesis, adipogenesis, and insulin sensitivity *in vitro* and *in vivo* (Donnelly *et al*, 2004; Yamawaki *et al*, 2005; Chen *et al*, 2008; Chutkow *et al*, 2008, 2010; Yoshioka *et al*, 2012). The molecular mechanisms responsible for the Txnip-null metabolic phenotype are not yet clearly defined. Of note, Txnip is one of the most dramatically upregulated genes in response to glucose in humans, suggesting a prominent role of Txnip in either adaptive or maladaptive changes of metabolism in response to glucose (Parikh *et al*, 2007).

Here, we report a new role of Txnip as a regulator of ER stress. We identified PDIs as a potentially critical mechanistic link between Txnip and UPR signaling. We show that Txnip binds to PDIs and increases PDI activity. Txnip deficiency leads to increased protein ubiquitination and UPR signaling, indicating that it might serve as a feedback regulator for diabetes-induced ER stress.

## Results

### Txnip interacts with PDIA6

Given the structural similarities between PDIs and thioredoxin, we hypothesized that Txnip interacts with PDIs and regulates their enzymatic activity. We established a pulldown assay that revealed a direct protein-protein interaction between Txnip and PDIA6 (Fig 1A). We have previously shown that Txnip interacts with thioredoxin covalently through a disulfide exchange reaction that requires the Txnip cysteine residue at position 247 (Patwari *et al*, 2006). We therefore tested whether this cysteine residue is also essential for interaction with PDIA6 using a C247S Txnip mutant. The C247S mutation abrogated Txnip's interaction with PDIA6 (Fig 1A). Next, we tested whether Txnip interacts with the redox active sites of PDIA6. PDIA6 contains two thioredoxin folds, each with a CGHC redox active motif (Hatahet & Ruddock, 2009). Pulldown analyses using full-length PDIA6 (1-421), the N-terminal domain (1-118), and the C-terminal domain (135-421) of PDIA6 showed that Txnip interacts with both the N- and the C-terminal thioredoxin folds of PDIA6 (Fig 1B). The N-terminal cysteine residue of CXXC redox active motifs is necessary for the initiation of disulfide exchange reactions through bimolecular nucleophilic substitution reactions (Lee *et al*, 2013). The reaction starts with the formation of a mixed disulfide bond following a nucleophilic attack of the N-terminal cysteine on the disulfide bond of the substrate protein. Subsequently, the intermediate mixed disulfide bond is targeted by a nucleophilic attack of the C-terminal cysteine, thereby completing the final disulfide exchange reaction (Lee *et al*, 2013). In order to test the hypothesis that the CGHC redox active motifs of PDIA6 are the sites of interaction with Txnip, we mutated cysteines in PDIA6 at positions 36 and 171 to serines, the N-terminal cysteine residues of the PDIA6 thioredoxin-fold CGHC motifs. Neither the C36S mutant nor the C171S mutants of PDIA6 bound to Txnip (Fig 1C and D). Mutation of the C-terminal resolving cysteine residue of CXXC motifs does not lead to abrogation of interaction, but rather in potential trapping of the substrate, more pronounced with cysteine to alanine mutations than with cysteine to serine mutation (Lee *et al*, 2013). We therefore mutated the

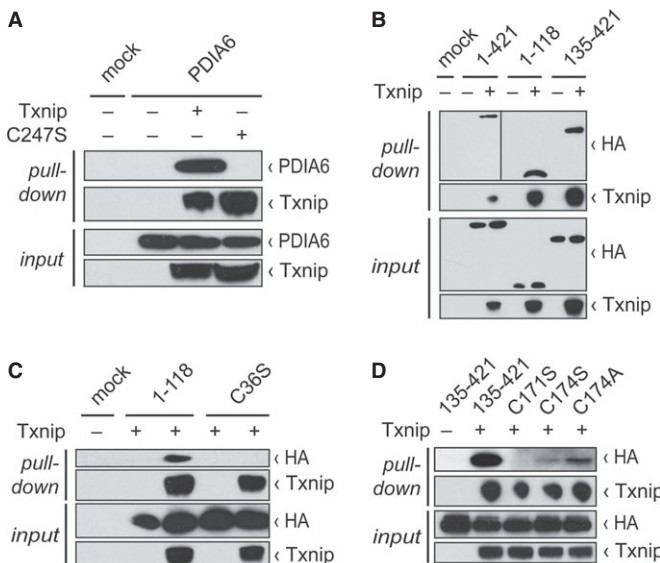

**Figure 1.   Txnip interacts with PDIA6.**

HEK293TN cells were transfected with the indicated plasmids, and Western blot analyses with the indicated antibodies of input lysates and pulldown eluates after affinity chromatography of Txnip were performed.

A   Pulldown of PDIA6 with Txnip but not with Txnip C247S mutant.

B   Pulldown of full-length PDIA6 (1-421), N-terminal (1-118), and C-terminal (135-421) domain of PDIA6 with Txnip.

C   Pulldown of wild-type N-terminal (1-118) domain of PDIA6, but not of N-terminal C36S mutant domain of PDIA6 with Txnip.

D   Pulldown of wild-type C-terminal (135-421) domain of PDIA6, but not of C-terminal C171S mutant of PDIA6, with Txnip. C174S and C174A mutations do not abrogate interaction with Txnip.

Source data are available online for this figure.

C-terminal cysteine residue of PDIA6 at position 174 to alanine and serine; as expected, the cysteine to alanine and serine mutants were still able to interact with Txnip (Fig 1D). These results show that PDIs with a CXXC redox motif bind to Txnip requiring the PDI redox active site, suggesting the possibility that Txnip can regulate PDI activity.

### Txnip is the only alpha arrestin interacting with PDIs

To test the specificity of the Txnip-PDI interaction, we tested the interactions of PDI, PDIA3, PDIA4, PDIA13, and PDIA15 with Txnip. Txnip bound to all of these CGHC-containing PDIs, but not with PDIA8 and PDIA9, which are PDIs that do not contain a CXXC redox active motif (Fig 2A). To further confirm these findings, we performed reverse pulldown analyses, which confirmed the interaction between PDIs and Txnip (Fig 2B). Txnip is one of six members of the arrestin domain-containing proteins sometimes called the alpha arrestins (Lee *et al*, 2013); since Txnip is the only arrestin domain-containing protein that interacts with thioredoxin, we tested whether other alpha arrestins bind to PDIA6. These experiments showed that Txnip is the only alpha arrestin that bound to PDIA6 (Fig 2C). Thus, Txnip interacts with a number CGHC-containing PDIs, a specific feature of Txnip not shared by other members of the alpha arrestin protein family.

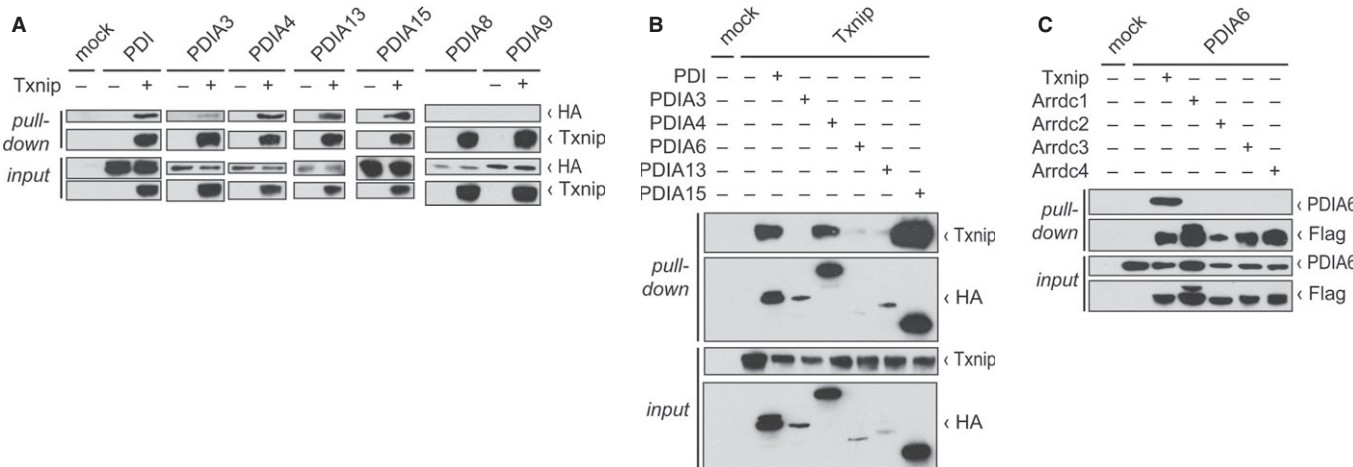

**Figure 2. Txnip is the only alpha arrestin that interacts with PDIs.**

HEK293TN cells were transfected with the indicated plasmids, and Western blot analyses with the indicated antibodies of input lysates and pulldown eluates after affinity chromatography were performed.

A  Pulldown of PDI, PDIA3, PDIA4, PDIA13, PDIA15, but not of PDIA8 and PDIA8 with Txnip.
B  Pulldown of Txnip with PDI, PDIA4, PDIA6, PDIA13, and PDIA15.
C  Pulldown of PDIA6 with Txnip but not with other arrestin domain-containing proteins Arrdc1-4.

Source data are available online for this figure.

## Txnip is located in the ER and interacts with endogenous PDIA6

Since PDIs are predominantly located in the ER (Hatahet & Ruddock, 2009), we investigated whether Txnip is also located in this cellular compartment. Previous studies have shown that addition of an artificial N-glycosylation site leads to partial (GT1.4) or subtotal (GT1.4tail) glycosylation of proteins that are localized in the luminal site of the ER (Kaup et al, 2011). We generated Txnip-GT1.4 and Txnip-GT1.4tail mutant constructs and performed Western blot analyses showing glycosylation of Txinp (Fig 3A), indicating that Txnip is located in the ER at some point of its life cycle. An important potential limitation of our pulldown assay is the use of PDI constructs with an N-terminal HA-tag which could interfere with intracellular trafficking. Therefore, we performed immunofluorescence analyses to confirm that HA-tagged PDI is still located in the ER (Supplementary Fig S1A and B). To test the hypothesis that Txnip interacts with PDIs in physiologic conditions, we first performed pulldown analyses in HEK293TN cells that were transfected with Txnip only, showing that Txnip interacts with endogenous PDIA6 (Fig 3B). To confirm these findings, we chose an unbiased proteomics approach to identify protein-protein interaction partners for Txnip (Gao et al, 2009). Affinity chromatography with subsequent SDS–PAGE and mass spectrometry analyses confirmed that Txnip interacts with several endogenously expressed PDIs (Table 1, Supplementary Fig S2 and see Supplementary Dataset S1). To assess the relative abundance of free Txnip versus Txnip that is in complex with PDI, we trapped Txnip-PDI complexes using a free sulfhydryl alkylation method (Chutkow & Lee, 2011). All free cysteine residues were blocked with N-ethylmaleimide (NEM); non-reducing SDS–PAGE and subsequent Western blot analyses showed that a small but significant amount of Txnip is complexed with PDI (Supplementary Fig S3A and B). Taken together, these results show that Txnip interacts with endogenous PDIA6.

## Txnip increases PDI activity

Since PDIs function primarily as oxidants, introducing disulfide bonds into folding proteins, their reduction potential is higher than that of denatured proteins (Hatahet & Ruddock, 2009). We therefore hypothesized that Txnip shifts the PDI redox equilibrium to the reduced state, thereby increasing its capacity to reduce other proteins. We measured PDI activity in the protein extracts from cells overexpressing PDI with and without Txnip using a coupled insulin reduction assay. In this assay, reduction of insulin catalyzed by PDI is coupled to the oxidation of $NADPH + H^+$, which is quantified by the decrease in absorbance at 340 nm. As expected, Txnip increased PDI activity in this assay (Fig 4A). To confirm these findings, we performed an additional PDI activity assay; the insulin turbidity

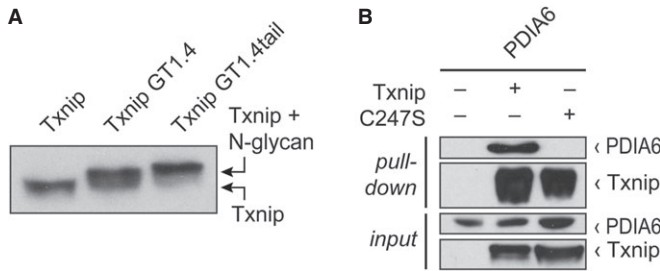

**Figure 3. Txnip is located in the ER and interacts with endogenous PDIA6.**

A  Western blot analysis of Txnip and Txnip mutants with an N-glycosylation site (Txnip GT1.4, and Txnip GT1.4tail) transfected into HEK293TN cells.
B  HEK293TN cells were transfected with Txnip or Txnip C247S mutant plasmids. Pulldown of endogenous PDIA6 with Txnip but not with Txnip C247S mutant.

Source data are available online for this figure.

**Table 1. Mass spectrometry results for Txnip protein-protein interactions**

| Protein | MW (kDa) | Control (hits) | Txnip (hits) | |
|---|---|---|---|---|
| Txnip | 44 | – | 2302 | Validation |
| Txn | 12 | – | 388 | |
| Txn2 | 18 | – | 16 | |
| Pdia4 | 73 | – | 72 | Protein disulfide isomerases |
| Pdia6 | 48 | – | 25 | |
| Pdia15 | 44 | – | 3 | |

assay is based on the aggregation of the B chain of insulin at increasing concentrations following reduction of insulin catalyzed by PDI, which is quantified by an increase in absorbance at 650 nm. The results of this experiment confirmed our previous finding, showing that Txnip increases PDI activity (Fig 4B). Taken together, these results reveal Txnip as a novel regulator of PDI function.

Accumulation of misfolded or unfolded proteins that are potentially toxic poses a threat to cellular integrity. The increase in protein misfolding that is associated with decreases in PDI activity induces ER stress, which triggers the UPR and defenses against ER stress (Uehara *et al*, 2006; Hotamisligil, 2010). A recent study confirmed that Xbp1s induces PDI expression as part of ER stress-related regulation of lipid homeostasis (Wang *et al*, 2012). Therefore, we performed immunofluorescence analyses of mouse embryonic fibroblasts (MEFs) from WT and Txnip-KO mice. Both at baseline and under tunicamycin-induced ER stress conditions, Txnip-KO MEFs had higher levels of PDI expression (Supplementary Fig S4A and B). This indicates that reduced PDI activity and increased ER stress could activate the UPR in Txnip-KO cells, ultimately leading to a compensatory increase in PDI levels.

### Txnip deficiency increases protein ubiquitination

Another adaptive cellular mechanism to ER stress is targeting misfolded or unfolded proteins for proteasomal degradation, a process known as endoplasmic reticulum-associated degradation (ERAD) (Guerriero & Brodsky, 2012). A key step in this process is

substrate ubiquitination that targets ER proteins for retrotranslocation to the cytosol and subsequent proteasomal degradation (Nakatsukasa *et al*, 2008). We hypothesized that changes in PDI chaperone and enzymatic activity could lead to the accumulation of unfolded and ubiquitinated proteins (Uehara *et al*, 2006). We analyzed the accumulation of ubiquitinated proteins in WT and Txnip-KO MEFs with and without subjecting them to ER stress through tunicamycin (Fig 5A, negative controls in Supplementary Fig S4C). Both at baseline and under stimulated conditions, the number of ubiquitin-positive accumulations of proteins markedly increased in a dose-dependent manner in the Txnip-KO cells, consistent with an increase in proteins targeted for degradation (Fig 5B). These findings suggest the possibility of increased UPR signaling in Txnip-KO cells.

### Txnip regulates ER stress

The best-studied and most conserved branch of the UPR is mediated through the inositol-requiring enzyme 1α (IRE1α) pathway (Hetz *et al*, 2011). IRE1α is a transmembrane kinase and endonuclease that directly binds to unfolded proteins, which leads to lateral oligomerization, autophosphorylation, and activation of its ribonuclease domain (Sidrauski & Walter, 1997; Gardner & Walter, 2011). This domain cleaves 26 nucleotides out of the mRNA encoding for X-box binding protein 1 (Xbp1), which leads to a translational frame shift and the generation of the active transcription factor Xbp1s (Sidrauski & Walter, 1997). This non-conventional splicing event is UPR specific, and there are no other Xbp1 activation pathways known to date (Hotamisligil, 2010; Hetz *et al*, 2011). Xbp1s initiates a transcriptional program that upregulates a broad spectrum of proteins involved in maintaining ER homeostasis. We investigated the effects of Txnip deficiency on Xbp1s levels by treating MEFs from WT and Txnip-KO mice with tunicamycin. There was a dramatic increase in Xbp1s transcript levels in Txnip-KO cells compared to WT both at baseline and under stimulated conditions in a dose-dependent manner (Fig 6A). This was also true for L-azetidine carboxylic acid and thapsigargin, two other ER stress-inducing reagents with mechanisms different from tunicamycin (Supplementary Fig S5A and B). Since PDI and PDIA6 expression is induced via the UPR pathway (Hatahet & Ruddock, 2009), we also studied the effect of Txnip deficiency on Pdi and Pdia6

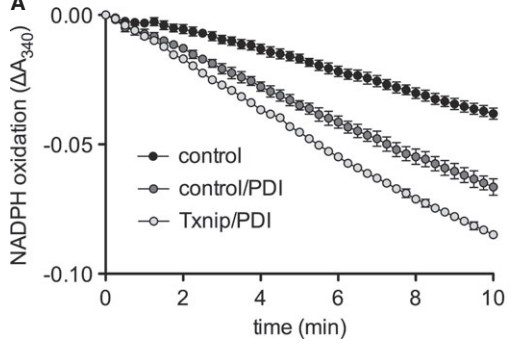
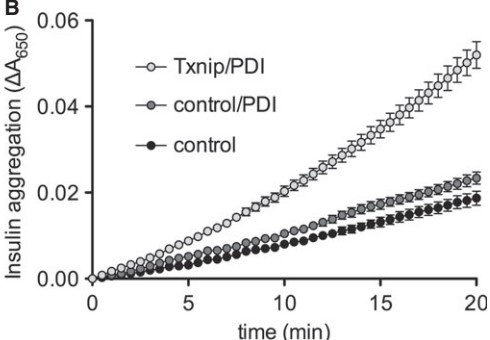

**Figure 4. Txnip increases PDI activity.**
HEK293TN cells were transfected with the indicated plasmids, and lysates were used to perform enzymatic activity assays.

A  PDI activity was measured using a coupled insulin reduction assay (*n* = 3).
B  PDI activity was measured using an insulin aggregation assay (*n* = 4).

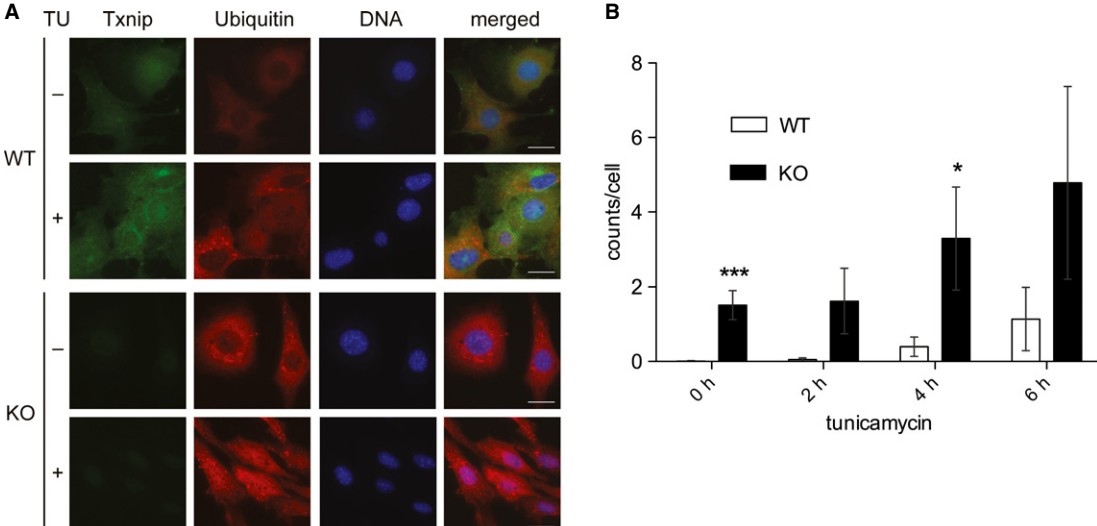

**Figure 5. Txnip deficiency increases protein ubiquitination.**

Mouse embryonic fibroblasts from wild-type (WT) and Txnip-null (KO) mice were treated with vehicle or tunicamycin (1 µg/ml) for 6 h. Cells were subsequently fixed, permeabilized, and stained for indicated proteins.

A    Levels of Txnip and mono- and polyubiquitinated protein accumulations visualized by immunofluorescence under epifluorescence microscopy. Scale bar, 25 µm.

B    Quantification of intracellular ubiquitin accumulations per field, normalized to cell number (n = 5 fields). *P = 0.04954, ***P = 0.00097 versus WT.

transcript levels in MEFs. Both at baseline and under ER stress, Pdi and Pdia6 transcript levels were significantly increased (Supplementary Fig S5C–E). Next, we investigated the reversibility of this effect in WT and Txnip-KO MEFs treated with tunicamycin by co-treatment with 4-phenylbutyric acid (PBA) and tauroursodeoxycholic acid (TUDCA). PBA and TUDCA are chemical chaperones that stabilize protein conformation, improve ER folding capacity, and facilitate trafficking of mutant proteins; as such, they have been shown to reduce ER stress levels (Ozcan *et al*, 2006). Co-treatment of tunicamycin-treated WT and Txnip-KO MEFs with PBA and TUDCA led to an almost complete normalization of Xbp1s transcript levels (Fig 6B). These results indicate that Txnip deficiency leads to a pronounced increase in splicing of Xbp1, revealing Txnip's role in controlling the UPR.

To investigate the effect of increased Txnip expression on Xbp1s levels, we subjected 3T3-L1 fibroblasts stably overexpressing Txnip to tunicamycin. There was a significant reduction in Xpb1s both at baseline and under stimulation in Txnip-overexpressing cells compared to WT (Fig 6C). To further study the role of PDI in this phenotype, we stably transduced Txnip-overexpressing 3T3-L1 cells with shRNA, knocking down PDI about 80% (PDI shRNA 3) and more than 95% (PDI shRNA 4, Txnip, and PDI expression levels of these cell lines in Supplementary Fig S6A and B). In addition, we treated Txnip-overexpressing cells with 16F16, a specific PDI inhibitor (Hoffstrom *et al*, 2010). While Txnip overexpression decreased Xbp1s transcript levels, this effect was reversed by knockdown of PDI in a transcript dose-dependent manner; pharmacologic inhibition of PDI had the same effect (Fig 6D). These data show that increased levels of Txnip significantly reduce Xbp1s transcript levels.

To confirm these findings at the protein level, we also performed Western blot analyses of Xbp1. Treatment of WT and Txnip-KO MEFs with tunicamycin led to an increase in Xbp1 expression in a dose-dependent manner at each time point in Txnip-KO MEFs compared to WT cells (Fig 6E). Co-treatment of these cells with the chemical chaperones PBA and TUDCA led to a reduction in Xbp1 protein levels compared to treatment with tunicamycin alone (Fig 6F). Txnip overexpression in 3T3-L1 fibroblasts led to a reduction in Xbp1 protein levels, both at baseline and under tunicamycin-stimulated conditions (Fig 6G).

To test whether Txnip also regulates UPR signaling *in vivo*, we extracted RNA from the livers of WT and Txnip-KO mice and measured transcript levels of UPR signaling molecules, including Xbp1s and its target genes Erdj3, Serp1, and Edem1. There was a robust increase in Xbp1s transcript levels as well as downstream targets of Xbp1s *in vivo* (Fig 7A–D). These changes in gene expression levels translated into increased protein levels of Xbp1 in the liver samples of Txnip-KO mice (Fig 7E). Next, we investigated whether these changes in ER stress signaling would also be reversible by the treatment with TUDCA and PBA *in vivo*. We generated liver-specific Txnip-KO mice and confirmed increased Xbp1 transcript levels compared to WT (Fig 7F). This increase was reversed by the treatment with chemical chaperones (Fig 7F). Taken together, these results reveal that Txnip is a regulator of Xbp1s levels and UPR signaling *in vivo*.

## Discussion

Beta and visual arrestins are well known as important regulators of G-protein-coupled receptor signaling. Canonically, they bind to active phosphorylated receptors and cause desensitization by mediating their ubiquitination and endosomal recycling (Lohse *et al*, 1990). More recently, the role of beta arrestins has expanded, as beta arrestins are now recognized to initiate their own signaling cascades independent of G-proteins as part of multifunctional signal-

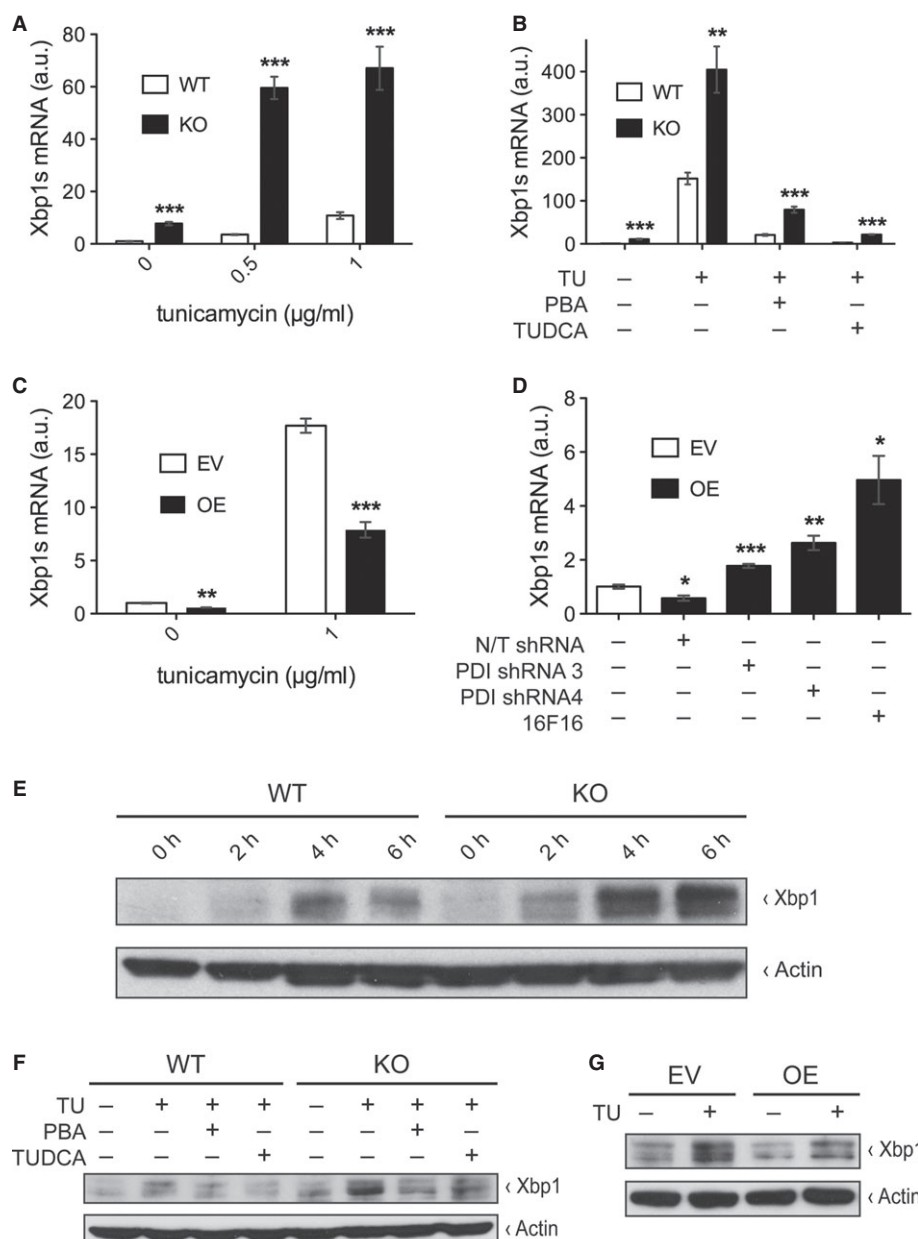

**Figure 6. Txnip regulates Xbp1s *in vitro*.**

A   Relative transcript levels of Xbp1s measured by qPCR normalized to 18S in mouse embryonic fibroblasts (MEFs) from wild-type (WT) and Txnip-null (KO) mice treated with increasing concentrations of tunicamycin for 2 h ($n = 4$). 0 μg/ml: ***$P$ = 0.00004 versus WT; 0.5 μg/ml: ***$P$ = 0.00001 versus WT; 1 μg/ml: ***$P$ = 0.0005 versus WT.

B   Same as in (A) in WT and KO MEFs treated with tunicamycin and chemical chaperones PBA (20 mM) and TUDCA (5 mg/ml) for 6 h ($n = 4$). TU/PBA/TUDCA −/−/−: ***$P$ = 0.00008 versus WT; TU/PBA/TUDCA +/−/−: **$P$ = 0.004 versus WT; TU/PBA/TUDCA +/+/−: ***$P$ = 0.0002 versus WT; TU/PBA/TUDCA +/−/+: ***$P$ = 0.0000003 versus WT.

C   Same as in (A) in empty vector-transduced (EV) and Txnip-overexpressing (OE) 3T3-L1 fibroblasts treated with tunicamycin for 2 h ($n = 4$). 0 μg/ml: **$P$ = 0.001 versus EV; 1 μg/ml: ***$P$ = 0.00007 versus EV.

D   Same as in (A) in EV and OE 3T3-L1 fibroblasts transduced with non-targeting (N/T) shRNA, PDI shRNA, or treated with PDI inhibitor 16F16 (5 μM) for 8 h ($n = 4$). +/−/−/−: *$P$ = 0.01 versus EV; −/+/−/−: ***$P$ = 0.00008 versus N/T shRNA; −/−/+/−: **$P$ = 0.002 versus N/T shRNA; −/−/−/+: *$P$ = 0.016 versus N/T shRNA.

E-G   Protein levels of Xbp1 and actin measured by Western blot analyses in: (E) WT and KO MEFs at increasing durations of treatment with tunicamycin (1 μg/ml), (F) WT and KO MEFs after treatment with tunicamycin (1 μg/ml for 2 h) and chemical chaperones PBA (20 mM) and TUDCA (5 mg/ml), and (G) EV and Txnip 3T3-L1 fibroblasts with and without tunicamycin treatment (1 μg/ml for 2 h).

Source data are available online for this figure.

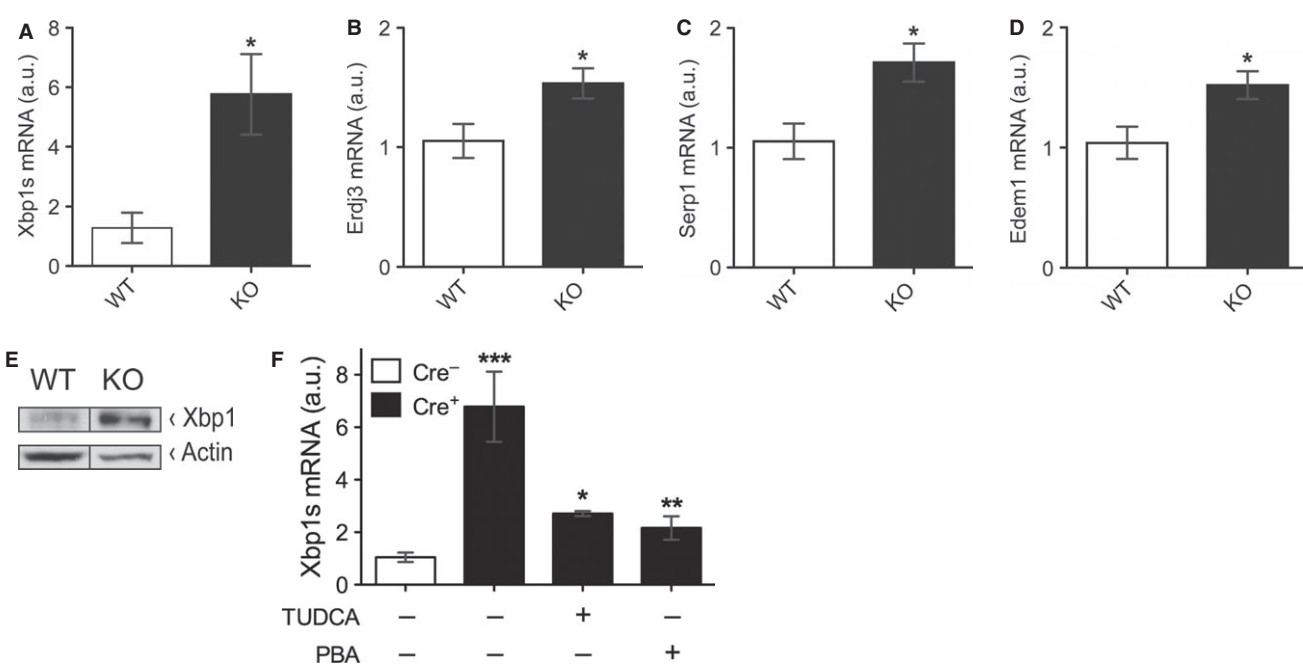

**Figure 7. Txnip regulates Xbp1s *in vivo*.**

A–D  Relative transcript levels of Xbp1s, Erdj3, Serp1, and Edem1 measured by qPCR normalized to 18S from liver samples from wild-type (WT) and Txnip-null (KO) mice
($n = 5$), [*$P = 0.04$ versus WT in (A); *$P = 0.02$ versus WT in (B); *$P = 0.04$ versus WT in (C), *$P = 0.03$ versus WT in (D)].

E  Protein levels of Xbp1 and actin measured by Western blot analyses in liver samples of WT and KO mice.

F  Relative transcript levels of Xbp1s in liver samples of liver-specific Txnip-KO mice (Cre[+]) and littermate controls (Cre[−]) treated with vehicle, tauroursodeoxycholic
acid (TUDCA), or 4-phenylbutyric acid (PBA) for 21 days ($n = 3–4$). *$P = 0.019$, **$P = 0.005$ versus Cre[+] (vehicle only); ***$P = 0.001$ versus Cre[−] (vehicle only).

Source data are available online for this figure.

ing scaffolds (Rajagopal *et al*, 2010). Phylogenetic studies have shown that beta arrestins are members of a larger arrestin superfamily with predicted structural similarities, including the two arrestin folds that identify arrestins in general (Alvarez, 2008). The proteins sometimes called alpha arrestins are phylogenetically more ancient than beta arrestins and include Txnip and arrestin domain-containing protein 1-5 (Arrdc1-5). Accumulating evidence suggests that alpha arrestins, similar to the extensively studied beta arrestins, participate in a variety of signaling pathways, establishing arrestin domain-containing proteins as integrators of signaling pathways (Patwari *et al*, 2009, 2011; Nabhan *et al*, 2012). For example, we have recently shown that Arrdc3 regulates obesity and energy expenditure in adipose tissue through interaction with β-adrenergic receptors in humans and mice (Patwari *et al*, 2011).

As part of the alpha arrestin family, Txnip also shares the arrestin-like sequence homology with other arrestin domain-containing proteins, and indeed, it inhibits cellular glucose uptake independent of its ability to bind to thioredoxin (Patwari *et al*, 2009). However, here we report that Txnip's unique capability among the alpha arrestins to interact with thioredoxin-like proteins, such as PDIs, allows it to participate in both redox-dependent and redox-independent metabolic signaling pathways. We show that Txnip interacts with PDIs and increases PDI activity. As a consequence, we identified Txnip as a novel regulator of ER stress and UPR signaling including Xbp1s.

Recent reports showed that Txnip itself is induced by ER stress, serving as a link between ER stress and inflammatory signaling

(Lerner *et al*, 2012; Oslowski *et al*, 2012). Activation of IRE1α leads to the reduction in the Txnip mRNA destabilizing microRNA-17 (Lerner *et al*, 2012). This results in increased levels of both Txnip mRNA and Txnip protein upon stimulation with different ER stress-inducing agents. Txnip subsequently activates the NLRP3 inflammasome, which results in increased interleukin 1β secretion (Zhou *et al*, 2010). These studies indicate that Txnip is an ER stress-sensitive gene that is involved in linking UPR signaling with inflammatory activation. While increased ER stress levels induce Txnip expression, our results show that increased Txnip levels ultimately lead to decreased Xbp1s levels, the downstream signaling molecule of IRE1α. Taken together, these data reveal a negative feedback mechanism by which increased IRE1α activation leads to the induction of Txnip, which subsequently results in increased PDI activity and downregulation of IRE1α signaling. Since Txnip is one of the most strongly upregulated genes in diabetes (Parikh *et al*, 2007), this reveals Txnip as a key negative feedback regulatory mechanism of the UPR to decrease Xbp1s levels via direct regulation of PDI.

Being under post-transcriptional regulation by the IRE1α pathway and affecting its downstream signaling molecule Xbp1s suggests that Txnip is involved in downstream signaling of this pathway as well. Several studies have shown that Xbp1s decreases gluconeogenesis and increases lipogenesis, adipogenesis, and insulin sensitivity (Ozcan *et al*, 2004; Lee *et al*, 2008; Sha *et al*, 2009; Zhou *et al*, 2011). Interestingly, Txnip deficiency in mice is associated with decreased gluconeogenesis, increased lipogenesis,

adipogenesis, and insulin sensitivity—similar to the Xbp1s metabolic phenotype (Donnelly *et al*, 2004; Chutkow *et al*, 2008, 2010). The present study therefore provides evidence that some of the metabolic phenotypes of Txnip deficiency may be attributable to its regulation of UPR signaling.

It is somewhat surprising that Txnip, which has been characterized as an inhibitor of thioredoxin activity, increases activity of the thioredoxin fold-containing PDI. However, PDI primarily functions as an oxidase and has a higher standard reduction potential ($E_r^{0\prime} = -180$ mV) than denatured proteins ($E_r^{0\prime} = -220$ mV), while thioredoxin that functions as a reductase has a lower standard reduction potential ($E_r^{0\prime} = -270$ mV) (Krause *et al*, 1991; Lundstrom & Holmgren, 1993; Hatahet & Ruddock, 2009). We therefore hypothesized that Txnip, being oxidized by PDI, shifts the PDI redox equilibrium to the reduced state, thereby increasing its capacity to reduce other proteins, which was confirmed in our *in vitro* PDI activity assays. Subjecting cells to ER stress increases PDI expression levels (Hatahet & Ruddock, 2009); but given the complex nature of the UPR, it is not surprising that overall PDI activity decreased after subjecting cells to ER stress using tunicamycin (Supplementary Fig S7). These findings highlight a limitation of the present study, which is the use of crude cell extracts for PDI activity assays, which also contain additional proteins like thioredoxin reductase and thioredoxin as well as glutaredoxins, which could interfere in the assay. Ideally, purified proteins should be used in enzymatic assays. However, despite efforts by multiple laboratories including our own to purify Txnip protein (Supplementary Figs S8 and S9), only crystallization of the N-terminal domain of human Txnip has been published so far (Polekhina *et al*, 2011). We have previously shown that results from enzymatic assays using crude cell extracts correlate well with *in vitro* assays using purified proteins (Holmgren & Bjornstedt, 1995; Patwari *et al*, 2006).

In conclusion, we report Txnip as a new regulator of ER stress through a direct interaction with PDIs. Txnip deficiency results in increased protein ubiquitination and increased Xbp1s signaling *in vitro* and *in vivo*, indicating that Txnip might serve as a feedback regulator for diabetes-induced ER stress.

# Materials and Methods

## Reagents

All reagents for biochemical and molecular biology assays were obtained from Sigma-Aldrich unless indicated otherwise. All reagents for cell culture experiments were obtained from Invitrogen unless indicated otherwise. Anti-Txnip antibody for Western blot analysis was from MBL International (JY2), anti-VDUP1 antibody for immunofluorescence was from Invitrogen, anti-PDIA6 antibody was from Abnova (3B4), anti-FLAG antibody was from Sigma-Aldrich (M2), anti-HA antibody was from Covance (16B12), anti-PDI antibody was from Novus Biologicals (RL77), anti-ubiquitin antibody was from Enzo Life Sciences (FK2), anti-Xbp1 antibody was from Santa Cruz, and anti-actin antibody was from Sigma-Aldrich. 4-Phenylbutyric acid and tauroursodeoxycholic acid were obtained from EMD Chemicals.

## Plasmid construction

Lentiviral vectors for transient transfection and stable transduction were used as described previously (Patwari *et al*, 2011; Yoshioka *et al*, 2012). Briefly, Txnip, Txnip C247S, and ARRDC1-4 were subcloned into pCDH-CMV-MCS-EF1-Green-T2A-Puro (System Biosciences) with an N-terminal tandem Strep/FLAG epitope tag (mDYKDDDDKgsaasWSHPQFEKgggsgggsgggsWSHPQFEK) (Gloeckner *et al*, 2009). PDI, PDIA3, PDIA4, PDIA6, PDIA6 (1-118), PDIA6 (1-118) C36S, PDIA6 (135-421), PDIA6 (135-421) C171S, PDIA6 (135-421) C174S, PDIA6 (135-421) C174A, PDIA8, PDIA9, PDIA13, and PDIA15 were subcloned into pCDH-CMV-MCS-EF1-Green-T2A-Puro (System Biosciences) with an HA epitope tag at the N-terminus of the signal sequence (YPYDVPDYA). cDNAs for ARRDC1, ARRDC2, PDI, PDIA3, PDIA4, PDIA8, PDIA9, and PDIA13 were subcloned from commercially available plasmids (Open Biosystems). PDIA6 (1-118), PDIA6 (1-118) C36S, and PDIA6 (135-421) constructs were kind gifts from Dr Roland K. Strong (Fred Hutchinson Cancer Research Center, Seattle, WA). PDIA6 (135-421) C171S, PDIA6 (135-421) C174S, and PDIA6 (135-421) C174A mutations were made using 'Round the Horn' site-directed mutagenesis as described previously (Chutkow & Lee, 2011). Txnip-GT1.4 and Txnip-GT1.4tail mutants were generated through the addition of C-terminal artificial N-glycosylation sites (GT1.4: LEAAAAAA<u>NAT</u>V, GT1.4tail: LEAAAAAA<u>NAT</u>VAAASGDVWDI), as described previously (Kaup *et al*, 2011) and subsequent subcloning into pCDH-CMV-MCS-EF1-Green-T2A-Puro (System Biosciences). PDI shRNA constructs were made using the following oligonucleotides (TRC shRNA Library):

mPdi shRNA #1 F: 5′-CCGGGCATTTCATCTGTGAGGCATTCTC-GAGAATGCCTCACAGATGAAATGCTTTTTG-3′, mPdi shRNA #1 R: 5′-AATTCAAAAAGCATTTCATCTGTGAGGCATTCTCGAGAATGCCT CACAGATGAAATGC-3′, mPdi shRNA #2 F: 5′-CCGGGCAGAGGC-TATTGATGACATACTCGAGTATGTCATCAATAGCCTCTGCTTTTTG -3′, mPdi shRNA #2 R: 5′-AATTCAAAAAGCAGAGGCTATTGATGA-CATACTCGAGTATGTCATCAATAGCCTCTGC-3′, mPdi shRNA #3 F: 5′-CCGGCCCAAGAGTGTATCTGACTATCTCGAGATAGTCAGATACA CTCTTGGGTTTTTG-3′, mPdi shRNA #3 R: 5′-AATTCAAAAACC-CAAGAGTGTATCTGACTATCTCGAGATAGTCAGATACACTCTTGG G-3′, mPdi shRNA #4 F: 5′-CCGGGCTCTGAGATTCGACTAG-CAACTCGAGTTGCTAGTCGAATCTCAGAGCTTTTTG-3′, mPdi shRNA #4 R: 5′-AATTCAAAAAGCTCTGAGATTCGACTAGCAACTCGAGTTG CTAGTCGAATCTCAGAGC-3′, mPdi shRNA #5 F: 5′-CCGGCAGCG-CATACTTGAGTTCTTTCTCGAGAAAGAACTCAAGTATGCGCTGTTT TTG-3′, mPdi shRNA #5 R: 5′-AATTCAAAAACAGCGCATACTT-GAGTTCTTTCTCGAGAAAGAACTCAAGTATGCGCTG-3′, N/T shRN A F: 5′-CCGGCAACAAGATGAAGAGCACCAACTCGAGTTGGTGCTC TTCATCTTGTTGTTTTTG-3′, N/T shRNA R: 5′-AATTCAAAAACAA-CAAGATGAAGAGCACCAACTCGAGTTGGTGCTCTTCATCTTGTTG-3′. Oligos were inserted into a pLKO.1 neo vector (Addgene #13425 contributed by Sheila Stewart) and stably transduced into the indicated cell lines as described previously (Patwari *et al*, 2006).

## Pulldown assay

Pulldown assays were performed as previously described (Yoshioka *et al*, 2012). Briefly, indicated plasmids were transfected into

HEK2937TN cells using PureFection transfection reagent (System Biosciences). Cells were lysed in 0.5% Triton X-100, 150 mM NaCl, 50 mM Tris, 1 mM phenylmethanesulfonyl fluoride, and protease inhibitors, pH 7.8. Txnip pulldown assay was performed by affinity chromatography using magnetic Streptactin beads (IBA) according to the manufacturer's instructions using a wash buffer containing 0.5% Triton X-100, 500 mM NaCl, 50 mM Tris, pH 7.8. Input lysates and pulldown eluates were analyzed by SDS–PAGE and immunoblots.

### Proteomics screen for Txnip protein-protein interactions

Txnip was subcloned into pCDH-CMV-MCS-EF1-Puro (System Biosciences) with an N-terminal tandem Strep/FLAG epitope tag (mDYKDDDDKgsaasWSHPQFEKgggsgggsgggsWSHPQFEK) (Gloeckner et al, 2009). HEK293F cells were stably transduced with Txnip or an empty vector control and clonally selected; cell culture was scaled up in suspension cell culture (Integra). Affinity chromatography was performed using Streptactin resin columns (IBA) according to the manufacturer's instructions. Eluates were subjected to SDS–PAGE and mass spectrometry analysis as described previously (Gao et al, 2009; Yoshioka et al, 2012).

### Coupled insulin reduction assay

PDI activity was assayed as described previously (Lambert & Freedman, 1983). In this assay, PDI uses reduced glutathione (GSH) as an electron donor for the reduction of insulin. This reaction is then coupled to a second redox reaction in which glutathione reductase catalyzes the reduction of oxidized glutathione (GSSG) using $NADPH + H^+$ as an electron donor. HEK293TN cells were transfected with the indicated plasmids; cell lysates were added to a reaction mix with the final concentrations of 30 μM insulin, 8 mM GSH, 1 U/ml glutathione reductase, and 120 μM NADPH in a buffer containing 100 mM $K_3PO_4$, 1 mM EDTA, pH 7.0. The oxidation of $NADPH + H^+$ to $NADP^+$ was monitored by the decrease in absorbance at 340 nm.

### Insulin turbidity assay

The assay was performed as described previously (Holmgren, 1979). In this assay, PDI breaks the two disulfide bonds between the insulin A and B chains, resulting in the precipitation of B chain. This precipitation can be monitored by an increase in absorbance at 650 nm. HEK293TN cells were transfected with the indicated plasmids; cell lysates and recombinant PDI were added to a reaction mix with the final concentrations of 30 μM insulin, 20 μg recombinant PDI, and 1 mM DTT, in a buffer containing 84 mM sodium phosphate, 2.67 mM sodium EDTA, pH 7.0. Precipitation of insulin was measured by an increase in absorbance at 650 nm.

### Immunofluorescence

Cells were plated on chambered coverglass (Thermo Scientific) coated with 1 μg/ml fibronectin. After fixation in 4% paraformaldehyde (Electron Microscopy Sciences), cells were permeabilized in ice-cold methanol at −20°C and blocked in 5% goat serum (Vector Laboratories). After incubation with the indicated primary antibodies and respective species-matched AlexaFluor-conjugated secondary antibodies (Invitrogen), and Hoechst, cells were imaged using a

Zeiss LSM 710 confocal, a Zeiss LSM 510 confocal, and a standard epifluorescence microscope. Ubiquitin-positive accumulations were quantified with ImageJ (NIH, Bethesda, MD) using a standardized threshold and normalized to cell count.

### Gene expression analysis

Gene expression analysis was performed as described previously (Chutkow et al, 2010). Briefly, RNA extraction from cells and tissues was performed using Trizol reagent (Invitrogen) according to the manufacturer's instructions. cDNA was obtained from 2 μg of total RNA using the TaqMan Reverse Transcription kit (Applied Biosystems). Quantitative real-time PCRs were performed in a 7300 Real-time PCR system (Applied Biosystems). Relative amounts of mRNA were normalized to 18S. Primer sequences were from the MGH Primer Bank (Spandidos et al, 2010) or described previously (Oslowski & Urano, 2011): Xpb1s F: 5′-CTGAGTCCGAATCAGGTG-CAG-3′, Xbp1s R: 5′-GTCCATGGGAAGATGTTCTGG-3′, Erdj3 F: 5′-GTACCTCATCGGGACTGTGAT-3′, Erdj3 R: 5′-CAGAACCTCATAAG-CAGCACC3′, Serp1 F: 5′-GCAACGTCGCTAAGACCTC-3′, Serp1 R: 5′-CATGCCCATCCTGATACTTTGAA-3′, Edem1 F: 5′-CTACCTGC-GAAGAGGCCG-3′, Edem1 R: 5′-CTACCTGCGAAGAGGCCG-3′, Pdi F: 5′-GCCGCAAAACTGAAGGCAG-3′, Pdi R: 5′-GGTAGCCACGGAC ACCATAC-3′, Pdia6 F: 5′-AGCTGCACCTTCTTTCTAGCA-3′, Pdia6 R: 5′-CAGGCCGTCACTCTGAATAAC-3′.

### Animals

Generation of transgenic Txnip[fl/fl] mice containing a loxP-flanked exon 1 locus was described previously (Yoshioka et al, 2007). Total Txnip-KO mice were obtained by crossing Txnip[fl/fl] mice with Prot-amine-Cre transgenic mice resulting in heterozygous total Txnip-KO mice. Intercrossing of heterozygous mice resulted in homozygous Txnip-KO mice; only mice without the Cre transgene were used for further breeding (Yoshioka et al, 2007, 2012; Chutkow et al, 2008, 2010). The following primers were used for Txnip-KO genotyping as described previously: F1: 5′-TTT CGT TTG GGT TTT CAA GC-3′, F2: 5′-CTT CAC CCC CCT AGA GTG AT-3′, R2: 5′-CCC AGA GCA CTT TCT TGG AC-3′ (Yoshioka et al, 2007). Liver-specific Txnip-KO mice were obtained by crossing Txnip[fl/fl] mice with Albumin-Cre trans-genic mice (Chutkow et al, 2008). The following primers were used for Cre genotyping as described previously: F: 5′-GCG GTC TGG CAG TAA AAA CTA TC-3′, R: 5′-GTG AAA CAG CAT TGC TGT CAC TT-3′ (Chutkow et al, 2008). Mice were housed under controlled tempera-ture and lighting with free access to water and food. Mice were main-tained and experiments with mice were performed in accordance with the Institutional Animal Care and Use Committees of Harvard Medical School. All animal experiments were approved by the Institutional Animal Care and Use Committees of Harvard Medical School. In vivo studies were performed as described previously with minor modifica-tions (Ozcan et al, 2006). Male 8- to 12-week-old mice received intraperitoneal injections of 250 mg/kg TUDCA twice a day (9 AM and 9 PM, 500 mg/kg/day) for 21 days. Control animals received intraperitoneal injections of vehicle at the same time points for the same period of time. PBA was given in the drinking water at a concentration of 20 mM for 21 days (Zode et al, 2011). Control animals were given vehicle in the drinking water for the same period of time.

## The paper explained

### Problem
From 1980 through 2011, the number of Americans with diabetes increased from 5.6 million to 25.8 million. Moreover, there are currently 79 million people in the United States who exhibit prediabetic symptoms. The epidemic of obesity and type 2 diabetes challenges us to gain deeper understanding of the underlying mechanisms that link supply of excess nutrients to these conditions. One mechanism that has emerged recently focuses on the endoplasmic reticulum, which is responsible for folding, modification, and trafficking of a large number of secreted and membrane proteins. Chronic dysregulation of the endoplasmic reticulum, also known as ER stress, is detrimental and leads to maladaptive changes in cellular signaling that ultimately contribute to several disease processes including β-cell dysfunction, insulin resistance, and diabetes. The molecular mechanisms that underlie these pathophysiological processes need to be identified.

### Results
Thioredoxin-interacting protein (Txnip) is one of the most dramatically upregulated genes in response to glucose, suggesting a prominent role of Txnip in either adaptive or maladaptive changes in metabolism in response to glucose. We identified a protein-protein interaction of Txnip with protein disulfide isomerases (PDIs), which are chaperones essential for protein folding in the endoplasmic reticulum. We found that Txnip increases PDI activity suggesting a regulatory role for protein folding. We hypothesized that deletion of Txnip would lead to decreased PDI activity and an increased amount of misfolded proteins. We found that deletion of Txnip leads to increased levels of ubiquitinated proteins that are targeted for degradation and to increased levels of ER stress signaling *in vitro* and *in vivo*.

### Impact
Our findings have identified Txnip as a novel regulator of PDI activity, protein folding, and ER stress. The results of this study add to our understanding of the underlying mechanisms that are relevant for the development of ER stress, a key mechanism for the development of insulin resistance and ER stress. This might help us find new therapeutic targets for the treatment of these diseases in the future.

For more detailed methods see the Supplementary Materials and Methods.

### Statistics

All data are presented as mean $\pm$ SEM with the number ($n$) of independent experiments underlying each data point. Differences between the groups were tested by unpaired two-tailed Student's *t*-test. Differences between multiple groups were tested by one-way ANOVA followed by *post hoc* analysis with the Tukey–Kramer test. A *P*-value <0.05 was considered statistically significant.

**Supplementary information** for this article is available online: http://embomolmed.embopress.org

### Acknowledgements
We thank Roland Strong for generously providing PDI reagents. We thank the CECAD Imaging Facility, University of Cologne, Germany, for providing excellent technical assistance. This work was supported by the Köln Fortune Program, Faculty of Medicine, University of Cologne, Germany (to SL), the German Research Foundation (LE 2728/1-1 to SL), the American Diabetes Association (7-12-MN-46 to RTL and SL), the American Heart Association (13GRNT1687007 to JY), and the National Institutes of Health (HL048743, HL103582).

## Author contributions
SL, WAC, and RTL designed research; SL, WAC, SMK, JD, LL, RBM, and EPF performed research; SL, EPF, SB, WAC, PP, JY, and RTL analyzed data; SL and RTL wrote the paper.

## Conflict of interest
The authors declare that they have no conflict of interest.

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

ER stress via a chemical chaperone prevents disease phenotypes in a
mouse model of primary open angle glaucoma. *J Clin Invest* 121:
3542−3553

