## [Review Process File · EMBO Molecular Medicine]

Thioredoxin-interacting protein regulates protein disulfide isomerases and endoplasmic reticulum stress

Samuel Lee, Soo Min Kim, James Dotimas, Letitia Li, Edward P. Feener, Stephan Baldus, Ronald B. Myers, William A. Chutkow, Parth Patwari, Jun Yoshioka, and Richard T. Lee

Corresponding author: Richard Lee, Brigham and Women's Hospital, Harvard Medical School

Review timeline:

Submission date:	27 January 2013
Editorial Decision:	01 February 2013
Appeal:	01 February 2013
Editorial Decision:	27 February 2013
Revision received:	07 October 2013
Editorial Decision:	28 October 2013
Revision received:	01 March 2014
Editorial Decision:	02 April 2014
Revision received:	17 April 2014
Accepted:	22 April 2014

Transaction Report:

Editor: Céline Carret

1st Editorial Decision

01 February 2013

Thank you for the submission of your manuscript "Thioredoxin-interacting protein regulates protein disulfide isomerases and endoplasmic reticulum stress".

Upon receipt, manuscripts are evaluated by the Scientific Editors to deal in a timely fashion with the large number of submissions that we receive. In this case, I am afraid that we concluded that the manuscript is not well suited for publication in EMBO Molecular Medicine and have therefore decided not to proceed with peer review.

The main reason for this decision is the uncertain clinical impact of your findings as no in vivo experimentation in disease animal models is provided. While we do understand that Txnip is here identified as a regulator of ER stress and ER stress was previously linked to diseases, without a clear pathophysiological significance of this finding, the study does not fit the scope of EMBO Molecular Medicine.

I am sorry that I could not bring better news this time.

Appeal

01 February 2013

Thank you for your careful consideration of our manuscript "Thioredoxin-interacting protein regulates protein disulfide isomerases and endoplasmic reticulum stress" (EMM-2013-02561).

We understand your concern about the uncertain clinical impact of our findings; however, we are wondering if you will reconsider based on the following information: Both animal as well as human studies have shown that Txnip is one of the most powerfully induced genes in diabetes. We and others have shown that Txnip plays a critical role in regulation of glucose uptake and energy metabolism in animals and humans. Furthermore, endoplasmic reticulum stress is emerging as a potential therapeutic target for diabetes with several clinical trials completed or ongoing (ClinicalTrials.gov ID: NCT00533559, NCT00771901, NCT01211015). Our study describes for the first time a mechanistic link between Txnip and ER stress. Due to the importance of new therapeutic targets for the treatment of diabetes, we believe that these findings are highly clinically relevant as they reveal a new mechanism that can be exploited therapeutically in the future.

Given the potential importance of Txnip in diabetes, and that our paper reveals a major new molecular mechanism, we would appreciate it very much if you would consider obtaining peer review of the manuscript. We understand that this may ultimately lead to a negative decision in any case, but we're optimistic that reviewers will see the potential importance.

Thank you very much for your time.

2nd Editorial Decision

27 February 2013

Thank you for the submission of your manuscript to EMBO Molecular Medicine. We have now heard back from the three referees whom we asked to evaluate your manuscript. Although the referees find the study to be of potential interest, they all raise a common concern about using a more physiological setting.

As you will see from the reports below, all three referees question the physiological relevance of Txnip binding to PDIs as the experimental system is somehow artificial; endogenous Txnip should be shown to bind PDIs. Referee #3 would also like to see more evidence showing that Txnip binding increases PDI activity, thereby providing some mechanism to better understand the biological process.

In our view, the suggested revisions would render the manuscript much more attractive and relevant for the broad readership of EMBO Molecular Medicine. We hope that you will be prepared to experimentally address the issues raised, with the understanding that the referees' concerns must be fully addressed and that acceptance of the manuscript would entail a second round of review. However, please note that that it is our journal's policy to allow only a single round of revision, and that acceptance or rejection of the manuscript will therefore depend on the completeness of your response and the satisfaction of the referees with it.

As you know, EMBO Molecular Medicine has a "scooping protection" policy, whereby similar findings that are published by others during review or revision are not a criterion for rejection. However, I do ask you to get in touch with us after three months if you have not completed your revision, to update us on the status. Please also contact us as soon as possible if similar work is published elsewhere.

I look forward to receiving your revised manuscript.

***** Reviewer's comments *****

Referee #1 (Remarks):

Txnip, a member of arrestin protein family, is one of the most strongly up-regulated proteins in diabetic patients. This protein can bind thioredoxin and modulate its activity. Recent studies showed that Txnip itself is induced by ER stress and serves as a link between ER stress and ER stress-mediated programmed cell death. PDIs, members of thioredoxin superfamily, are ER luminal proteins involved in the formation of disulfide bonds into proteins that are folding in this compartment. In this study, the authors investigated a potential role of Txnip in the regulation of PDIs in the ER. Based on their results, the authors claim that Txnip directly binds PDIs and regulates PDI activity, thereby decreasing the ER stress. Their findings are potentially interesting. However, there seem to be problems in their experimental design and interpretation of the results as will be described below.

1. The authors claim that Txnip binds PDIs and acts as a direct regulator of PDIs. Since PDIs are ER luminal proteins, Txnip also must be localized within the ER to directly influence the enzyme activity of PDIs. Is it already established that Txnip is located within the ER? Otherwise, it is essential to demonstrate that this protein is localized in the luminal side of the ER. This issue is particularly important because the primary sequence of Txnip does not have a conventional signal sequence or an ER-retrieval KDEL sequence. Addition of an artificial N-glycosylation site to Txnip, for example, may allow the authors to test whether the protein is localized within the ER.

2. The author should also confirm that the tagged versions of PDIs are indeed localized within the ER and that they are not in the cytosol because it could happen that PDIs that failed to be exported formed a disulfide-linked complex with Txnip in the cytosol. In addition, this reviewer has a concern that it is described on p13 that PDIs were subcloned into a vector with an N-terminal HA epitope tag. The authors should explain more clearly where is the HA tag. If the tag was fused to the N-terminus of the signal sequence of each PDI, it may have hindered the function of the signal sequence of the protein, causing the accumulation of a large amount of the precursor protein in the cytosol. If the complexes are formed in the cytosol, this reviewer feels that the evidence presented in this manuscript is too weak for the claim that Txnip directly binds PDIs to regulate their enzyme activity.

3. An artificial disulfide-linked complex can form even during sample preparation. This can happen especially when two proteins that form the complex have reactive cysteine residues and when their expression levels are high. Formation of such artificial complexes can be minimized by blocking all free cysteine residues using an alkylation reagent such as NEM before free cysteine residues react with disulfides in the sample. To expose all free cysteine residues for their efficient alkylation, the proteins can be denatured by treating live cells directly with 10% TCA, followed by wash with acetone and cysteine alkylation in an appropriate buffer containing SDS and NEM. Alternatively, the formation of an artificial disulfide-linked complex may be minimized by blocking reactive free cysteine residues before cell lysis by growing cells for some time in the presence of NEM. These methods can be found in published papers. Although the authors have added NEM to the cell lysis buffer, this reviewer feels that their method can be improved to minimize the formation of artificial complexes.

4. In vitro experiments presented in Figure 3 should be performed using purified Txnip and purified PDIs to demonstrate that Txnip can directly interact with PDIs and modulate their activity in vitro. The use of purified components will also allow the authors to answer a fundamental question whether Txnip acts stoichiometrically or catalytically to modulate the activity of PDIs.

5. In the supporting information Figure 1, the authors argue that they detected a disulfide-linked complex between Txnip and the endogenous PDIA6. To interpret this result, it is important again to know the cellular localization of Txnip. This is because, if a large amount of Txnip exists in the cytoplasm but not in the ER, it can be possible that the disulfide-linked complex between Txnip and PDIA6 was formed in the cytosol.

Minor points:

1) Page 5, line 8, "a C274S Txnip mutant" should be read as "a C247S Txnip mutant".

Referee #2 (Comments on Novelty/Model System):

The finding of Txnip as a novel feedback regulator of UPR signaling to decrease ER stress is interesting. This finding, together the recent reported role of Txnip in ER stress and inflammasome, expand our knowledge of the complex regulation of Txnip in ER stress. However, the lack of endogenous protein interaction is a major issue that should be resolved.

Referee #2 (Remarks):

The finding of Txnip as a novel feedback regulator of UPR signaling to decrease ER stress is interesting. This finding, together the recent reported role of Txnip in ER stress and inflammasome, expand our knowledge of the complex regulation of Txnip in ER stress.

Major comments:

1. Figure 1 and 2 revealed specific interaction of Txnip-PDI. These pull-down assays were done with exogenous over-expressed PDI and Txnip. The authors should perform experiments that demonstrate that this interaction occurs between endogenous Txnip and PDI, either by co-IP or co-localization approach?

2. Figure 4, 5 and 6 demonstrated that Txnip deficiency exacerbated UPR. In figure 5 and 6, a negative regulatory role Txnip on UPR-Xbp-1s activation was revealed. But it's known that Xbp-1s induces PDI expression as part of UPR. So it's good to know the expression level and activity of PDI under these conditions, to connect Txnip-PDI interaction and UPR more tightly.

3. Figure 6 demonstrated that metabolic phenotype of Txnip deficiency may be attributable to its regulation of UPR signaling. ER stress has been linked to lipogenesis and hepatic lipid accumulation, and administration of chemical chaperones can alleviate atherosclerotic lesions (Hotamisligil GS 2010). So can chemical chaperones diminish Txnip KO mice liver Xbp-1s activation?

4. The discussion is not very clear on the relationship between the negative feedback of TXNIP on UPR and the phenotype of the TXNIP KO mouse.

Referee #3 (Remarks):

In this paper the authors have studied protein disulfide isomerases (PDIs) and their interaction with the thioredoxin-interacting protein (Txnip). Txnip is known to be induced by glucose and bind to thioredoxin via a mixed disulfide formation involving Cys247 of Txnip and N-terminal active site Cys residue of Trx. The authors have shown that Txnip binds also to PDIs, which have two Trx domains with the sequence -CGHC-. They have convincingly shown that Txnip indeed forms a covalent complex with both the intact PDI and the isolated Trx domains of several PDIs. Through mutational analysis they show that it is the N-terminal active site Cys residue, which is known to be the enzymatically active by attacking disulfide Cys residue, which is the target of Txnip.

On p 6 they then go on to show that Txnip increases the activity of PDI. The argument being that Txnip would shift PDI redox equilibrium to the reduced state thereby increasing its capacity they reduce other proteins. It is unclear how the authors mechanistically would explain this. Their results in Fig 3 A and 3 B is using cell extracts either over-expressing PDI or also Txnip with an artificial assay with high glutathione (8 mM) and insulin in the presence of a large excess of glutathione reductase and NADPH. Under these conditions, the extract containing PDI + Txnip, somewhat higher activity than the one with only PDI. The problem with this assay is that it is done with crude extract of cells, which also contain additional proteins like thioredoxin reductase and thioredoxin as well as glutaredoxins, which could interfere in the assay. In order to show that Txnip increases PDI activity, the authors should use a refolding assay of a protein like ribonuclease preferentially with pure protein and understand that conditions whereby the activity of ribonuclease is measured. In those conditions they would be able to show that Txnip indeed increases catalytic activity of PDI. As now, it is hard to understand how the binding of Txnip, which inhibits Trx activity, by binding the same way to PDI at the active site increase its catalytic activity. In Fig 3 B they use an insulin turbidity assay using a very low concentration of DTT of 10 μ M (should not this be 10 mM?). Is this indeed correct to get insulin precipitation? Again, this is an indirect assay of PDI activity using crude extracts but working in the wrong direction namely reducing a protein.

Thus the authors, to convincingly show that PDI activity is increased, they should repeat the experiments with pure protein and using a refolding assay to demonstrate their point of increased analytic activity. The next step of course will be how to explain the binding of Txnip to PDI can lead to increased catalytic activity.

We thank the reviewers' for their time and insightful comments and critiques. We revised the manuscript according to the reviewers' suggestions, which significantly improved the manuscript. Below, please find a point-by-point response to the issues brought up by the reviewers.

Reviewer #1

Txnip, a member of arrestin protein family, is one of the most strongly up-regulated proteins in diabetic patients. This protein can bind thioredoxin and modulate its activity. Recent studies showed that Txnip itself is induced by ER stress and serves as a link between ER stress and ER stress-mediated programmed cell death. PDIs, members of thioredoxin superfamily, are ER luminal proteins involved in the formation of disulfide bonds into proteins that are folding in this compartment. In this study, the authors investigated a potential role of Txnip in the regulation of PDIs in the ER. Based on their results, the authors claim that Txnip directly binds PDIs and regulates PDI activity, thereby decreasing the ER stress. Their findings are potentially interesting. However, there seem to be problems in their experimental design and interpretation of the results as will be described below.

Comment: The authors claim that Txnip binds PDIs and acts as a direct regulator of PDIs. Since PDIs are ER luminal proteins, Txnip also must be localized within the ER to directly influence the enzyme activity of PDIs. Is it already established that Txnip is located within the ER? Otherwise, it is essential to demonstrate that this protein is localized in the luminal side of the

ER. This issue is particularly important because the primary sequence of Txnip does not have a conventional signal sequence or an ER-retrieval KDEL sequence. Addition of an artificial N-glycosylation site to Txnip, for example, may allow the authors to test whether the protein is localized within the ER.

Response: We thank the reviewer for raising this important issue. We performed additional experiments according to the reviewer's suggestions and added new Figures 3C and D.

p. 33, para. 3: C. Western analysis of Txnip and Txnip mutants with an N-glycosylation site (Txnip GT1.4, and Txnip GT1.4tail) transfected into HEK293TN cells. D. Western analysis of Txnip and Txnip GT1.4tail transfected into HEK293TN cells treated with vehicle or Peptide-N-Glycosidase F (PNGase F).

p. 7, para. 1: Since PDIs are predominantly located in the ER (Hatahet & Ruddock, 2009), we investigated whether Txnip is also located in this cellular compartment. Previous studies have shown that addition of an artificial N-glycosylation site leads to partial (GT1.4) or subtotal (GT1.4tail) glycosylation of proteins that are localized in the luminal site of the ER (Kaup et al, 2011). We generated Txnip-GT1.4 and Txnip-GT1.4tail mutant constructs and performed Western analyses showing glycosylation of

Txnip (Fig 3C), which was reversed after treatment with an endoglycosidase (Fig 3D), indicating that Txnip is located in the ER at some point of its life cycle.

p. 16, para. 1: *Peptide-N-Glycosidase F (PNGase F) was from New England Biolabs and was used according to the manufacturer's instructions.*

p. 17, para. 1: *Txnip-GT1.4 and Txnip-GT1.4tail mutants were generated through addition of C-terminal artificial N-glycosylation sites (GT1.4: LEAAAAANATV, GT1.4tail: LEAAAAANATVAAASGDVWDI), as described previously (Kaup et al, 2011), and subsequent subcloning into pCDH-CMV-MCS-EF1-Green-T2A-Puro (System Biosciences).*

Comment: The author should also confirm that the tagged versions of PDIs are indeed localized within the ER and that they are not in the cytosol because it could happen that PDIs that failed to be exported formed a disulfide-linked complex with Txnip in the cytosol. In addition, this reviewer has a concern that it is described on p13 that PDIs were subcloned into a vector with an N-terminal HA epitope tag. The authors should explain more clearly where is the HA tag. If the tag was fused to the N-terminus of the signal sequence of each PDI, it may have hindered the function of the signal sequence of the protein, causing the accumulation of a large amount of the precursor protein in the cytosol. If the complexes are formed in the cytosol, this reviewer feels that the evidence presented in this manuscript is too weak for the claim that Txnip directly binds PDIs to regulate their enzyme activity.

Response: We agree with the reviewer that tagging PDIs at the N-terminus could affect intracellular trafficking. We added new Figures 3A showing pulldown of endogenous WT PDIA6 with Txnip, previously in Supporting Information Figure S1. In addition, we performed additional co-immunoprecipitation experiments showing an interaction between endogenous WT PDIA6 and endogenous WT Txnip. We also added the suggested information about the exact location of the HA tag in our Methods section.

p. 33, para. 3: Figure 3. Txnip interacts with endogenous PDIA6. A. HEK293TN cells were transfected with Txnip or Txnip C247S mutant plasmids. Pull-down of endogenous PDIA6 with Txnip but not with Txnip C247S mutant. B. Co-immunoprecipitation of endogenous PDIA6 with endogenous Txnip in mouse embryonic fibroblasts isolated from wildtype (WT) and Txnip-null (KO) mice using an anti-Txnip antibody (JY2) or non-specific IgG antibodies.

p. 6, para. 3: Txnip interacts with endogenous PDIA6. To test the hypothesis that Txnip interacts with PDIs in physiologic conditions, we first performed pulldown analyses in HEK293TN cells that were transfected with Txnip only showing that Txnip interacts with endogenous PDIA6 (Fig 3A).

p. 7, para. 1: *We then investigated the interaction of endogenous Txnip with endogenous PDIA6 using wildtype (WT) mouse embryonic fibroblasts (MEFs)—a cell type that expresses Txnip at a higher level than HEK293TN cells at baseline (Chutkow et al, 2010); we used MEFs isolated from Txnip-null (KO) mice as a negative control. Co-immunoprecipitations using a Txnip-specific monoclonal antibody that was generated in our laboratory (Yoshioka et al, 2007) revealed that endogenous Txnip interacts with endogenous PDIA6 (Fig 3B).*

p. 16, para. 2: *PDI, PDIA3, PDIA4, PDIA6, PDIA6 (1-118), PDIA6 (1-118) C36S, PDIA6 (135-421), PDIA6 (135-421) C171S, PDIA6 (135-421) C174S, PDIA6 (135-421) C174A, PDIA8, PDIA9, PDIA13, and PDIA15 were subcloned into pCDH-CMV-MCS-EF1-Green-T2A-Puro (System Biosciences) with an HA epitope tag at the N-terminus of the signal sequence (YPYDVPDYA).*

p. 19, para. 2: *Co-immunoprecipitations were performed as previously described (Patwari et al, 2011). Briefly, mouse embryonic fibroblasts isolated from WT and Txnip-KO mice were lysed in buffer containing 0.5% Triton X-100, 150 mM NaCl, 50 mM Tris, 1 mM phenylmethanesulfonylfluoride, and a protease inhibitor cocktail (Sigma-Aldrich, St. Louis, MO). Immunoprecipitation was performed using magnetic protein G beads (Invitrogen) incubated with anti-Txnip (JY2) or mouse IgG antibodies (Sigma-Aldrich). Western analysis of bound proteins was performed with anti-VDUP1 antibody (Invitrogen) followed by HRP-conjugated anti-rabbit IgG VeriBlot antibody (Abcam).*

Comment: An artificial disulfide-linked complex can form even during sample preparation. This can happen especially when two proteins that form the complex have reactive cysteine residues and when their expression levels are high. Formation of such artificial complexes can be minimized by blocking all free cysteine residues using an alkylation reagent such as NEM before free cysteine residues react with disulfides in the sample. To expose all free cysteine residues for their efficient alkylation, the proteins can be denatured by treating live cells directly with 10% TCA, followed by wash with acetone and cysteine alkylation in an appropriate buffer containing SDS and NEM. Alternatively, the formation of an artificial disulfide-linked complex may be minimized by blocking reactive free cysteine residues before cell lysis by growing cells for some time in the presence of NEM. These methods can be found in published papers. Although the authors have added NEM to the cell lysis buffer, this reviewer feels that their method can be improved to minimize the formation of artificial complexes.

Response: We thank the reviewer for these specific and helpful comments. We performed additional experiments according to the reviewer's suggestions and generated new Supporting Information Figure S1.

Supporting information p. 3, para. 1: Supporting Information Figure 1. Txnip interacts with endogenous PDI. HEK293TN cells were transfected with Txnip or Txnip C247S

mutant plasmids. Txnip-PDI complexes were resolved from free A. PDI and B. Txnip by non-reducing SDS-PAGE following NEM free sulfhydryl alkylation.

p. 7, para. 1: *All free cysteine residues were blocked with N-ethylmaleimide (NEM); non-reducing SDS-PAGE and subsequent Western analyses showed that a small but significant amount of Txnip is complexed with PDI (Supporting Information Fig S1A and B).*

Supporting Information p. 12, para. 1: *Free-sulfhydryls alkylation was performed as described previously (Chutkow & Lee, 2012). Briefly, 293TN cells were transfected with Txnip wild type and mutant constructs. 24 hours after transfection, cells were washed twice in ice-cold PBS, and proteins were precipitated with ice-cold 0.54 M trichloroacetate in PBS. Lysates were incubated on ice for 10 min, then centrifuged at 9000 x g for 10 min at 4 °C. The pellets were washed with acetone, then centrifuged at 9000 x g for 10 min at 4 °C. The resulting pellets were briefly air-dried on ice then dissolved in NEM-labeling buffer (62.5mM Tris, pH6.8, 1% SDS, 25mM NEM) and incubated at 4 °C with end-over-end rotation for 18 h, then incubated at 37 °C for 10 min. After adding nonreducing sample loading buffer and sonicating, the labeled proteins were subjected to SDS-PAGE and Western analysis.*

Comment: In vitro experiments presented in Figure 3 should be performed using purified Txnip and purified PDIs to demonstrate that Txnip can directly interact with PDIs and modulate their activity in vitro. The use of purified components will also allow the authors to answer a

fundamental question whether Txnip acts stoichiometrically or catalytically to modulate the activity of PDIs.

Response: We agree with the reviewer that enzymatic assays should ideally be done using purified proteins. We performed numerous extensive experiments to purify Txnip using different approaches, including expression in *E. coli* or baculovirus-infected insect cells. However, we were only able to purify Txnip in its denatured form, which is not suitable for enzymatic assays. We added some of the data to the new Supporting Information Figures S6 and S7. Purification, and ideally crystallization, of Txnip protein is a major goal in the arrestin field, which has eluded a number of laboratories so far. Only crystallization of the N-terminal domain of human Txnip has been published so far (Polekhina et al, 2011). We have previously shown that results from enzymatic assays using crude cell extracts correlate well with in vitro assays using purified proteins (Holmgren & Bjornstedt, 1995; Patwari et al, 2006). Including mock transfected cells as a negative control allows for the most accurate assessment of Txnip function in enzymatic assays at this point in time. To add more evidence that Txnip increases PDI activity, we performed a rhodanese chaperone assay, which does not involve any disulfide exchange reactions. In this assay, Txnip increased PDI activity as well. We hope that the reviewer agrees with us that taken together we present a compelling case that Txnip increases PDI activity. We added a new section in our Discussion to highlight the limitations of the present study.

Supporting Information p. 8, para. 1: Supporting Information Figure 6. Txnip purification. *A.* Schematic representation of recombinant human Txnip (hTxnip) with an N-terminal 6xHistidine-tag (6xHis) in pTrcHis TOPO (Invitrogen). *B.* Coomassie staining and *C.* Western analysis of protein lysates from *E. coli* overexpressing hTxnip (pellet), from supernatant of *E. coli* cultures (supernatant), and purified Txnip in denaturing conditions. Recombinant hTxnip protein was purified in denatured condition; however, solubility was limited in the supernatant fraction. *D.* Schematic representation of recombinant human Txnip with an N-terminal 6xHis-tag in fusion with *E. coli* Thioredoxin (ecTrx) in pET-32a(+) (Novagen). This system was used to enhance the

production and solubility of hTxnip. E. Western analysis of protein levels of ecTrx-6xHis-hTxnip fusion protein in E. coli lysates after increasing durations of induction with Isopropyl β -D-1-thiogalactopyranoside (IPTG). F. Coomassie staining of ec-Trx-6xHis-hTxnip fusion protein after Ni-NTA-column purification and subsequent SDS-PAGE before and after thrombin cleavage with corresponding Western analyses using an anti-Txnip (JY2) and an anti-His-tag antibody. Although hTxnip protein was successfully purified in native condition with this method, there were multiple bands in Coomassie staining, indicating significant amounts of contamination by other proteins.

Supporting Information p. 11, para. 1: Supporting Information Figure 7. Txnip aggregation. **A.** Chromatogram of *ecTrx-6xHis-hTxnip* fusion protein after thrombin cleavage and gel filtration high performance liquid chromatography (HPLC). **B.** Coomassie staining of SDS-PAGE of fractions 5/6 (*hTxnip*) and 20 (*ecTrx*). Dynamic light scattering (DLS) demonstrated that the particle size of purified **C.** *6xHis-hTxnip* and **D.** *ecTrx-6xHis-hTxnip* proteins was over 286 MDa and 181 MDa even with maximal usage of DTT. Thus, after purifying *hTxnip* protein by HPLC, *hTxnip* (fraction 5 and 6) formed protein aggregates.

p. 34, para. 1: *C.* PDI activity was measured using a rhodanese aggregation assay ($n = 3$).

p. 8, para. 1: In addition to catalyzing disulfide exchange reactions, PDI also functions as a chaperone for proteins that do not contain any disulfide bonds, such as rhodanese (Hatahet & Ruddock, 2009). To investigate the effect of Txnip on PDI chaperone activity,

we performed a rhodanese aggregation assay in which aggregation of denatured rhodanese is monitored by increase in absorbance at 320 nm. We hypothesized that PDI would decrease aggregation, but surprisingly PDI led to an increase in aggregation of rhodanese, which was enhanced through addition of Txnip (Fig 4C). This is most likely due to the fact that we used protein extracts from cells overexpressing PDI and Txnip and will be discussed in more detail below.

***p. 20, para. 2:** The assay was performed as described previously (Song & Wang, 1995). In this assay, aggregation of denatured rhodanese induced by guanidine hydrochloride is measured. 90 μ M rhodanese was denatured with 6 M guanidine hydrochloride for 30 min at room temperature and then diluted with cell lysates of HEK 293TN cells transfected with the indicated plasmids in a buffer containing 30 mM Tris, 50 mM KCl, pH 7.2. Aggregation of rhodanese was measured by increase in absorbance at 320 nm.*

***p. 15, para. 1:** However, we were surprised to find that in our rhodanese chaperone assay PDI increased rhodanese aggregation. Again, Txnip increased PDI activity in this assay as well, but we expected PDI to decrease rhodanese aggregation as had been described previously using purified proteins (Song & Wang, 1995; Uehara et al, 2006). These findings highlight a limitation of the present study, which is the use of crude cell extracts for PDI activity assays, which also contain additional proteins like thioredoxin reductase and thioredoxin as well as glutaredoxins, which could interfere in the assay. Ideally, purified proteins should be used in enzymatic assays. However, despite efforts by multiple laboratories including our own to purify Txnip protein (Supporting Information*

Fig S6 and Supporting Information Fig S7), only crystallization of the N-terminal domain of human Txnip has been published so far (Polekhina et al, 2011). We have previously shown that results from enzymatic assays using crude cell extracts correlate well with in vitro assays using purified proteins (Holmgren & Bjornstedt, 1995; Patwari et al, 2006). Including mock transfected cells as a negative control allows for the most accurate assessment of Txnip function in enzymatic assays at this point in time.

Comment: In the supporting information Figure 1, the authors argue that they detected a disulfide-linked complex between Txnip and the endogenous PDIA6. To interpret this result, it is important again to know the cellular localization of Txnip. This is because, if a large amount of Txnip exists in the cytoplasm but not in the ER, it can be possible that the disulfide-linked complex between Txnip and PDIA6 was formed in the cytosol.

Response: As we showed in our responses to the previous comments, we performed additional experiments and included new data in the current version of the manuscript that show that Txnip is localized in the ER.

Comment: Page 5, line 8, "a C274S Txnip mutant" should be read as "a C247S Txnip mutant".

Response: We apologize for this typo and corrected the mistake.

p. 5, para. 1: We therefore tested whether this cysteine residue is also essential for interaction with PDIA6 using a C247S Txnip mutant.

Reviewer #2

The finding of Txnip as a novel feedback regulator of UPR signaling to decrease ER stress is interesting. This finding, together the recent reported role of Txnip in ER stress and inflammasome, expand our knowledge of the complex regulation of Txnip in ER stress.

Comment: Figure 1 and 2 revealed specific interaction of Txnip-PDI. These pull-down assays were done with exogenous over-expressed PDI and Txnip. The authors should perform experiments that demonstrate that this interaction occurs between endogenous Txnip and PDI, either by co-IP or co-localization approach?

Response: We thank the reviewer for this helpful suggestion and performed experiments accordingly. Co-immunoprecipitation experiments showed an interaction between endogenous WT PDIA6 and endogenous WT Txnip.

p. 33, para. 3: B. *Co-immunoprecipitation of endogenous PDIA6 with endogenous Txnip in mouse embryonic fibroblasts isolated from wildtype (WT) and Txnip-null (KO) mice using an anti-Txnip antibody (JY2) or non-specific IgG antibodies.*

p. 7, para. 1: *We then investigated the interaction of endogenous Txnip with endogenous PDIA6 using wildtype (WT) mouse embryonic fibroblasts (MEFs)—a cell type that expresses Txnip at a higher level than HEK293TN cells at baseline (Chutkow et al, 2010); we used MEFs isolated from Txnip-null (KO) mice as a negative control. Co-immunoprecipitations using a Txnip-specific monoclonal antibody that was generated in our laboratory (Yoshioka et al, 2007) revealed that endogenous Txnip interacts with endogenous PDIA6 (Fig 3B).*

p. 19, para. 2: *Co-immunoprecipitations were performed as previously described (Patwari et al, 2011). Briefly, mouse embryonic fibroblasts isolated from WT and Txnip-KO mice were lysed in buffer containing 0.5% Triton X-100, 150 mM NaCl, 50 mM Tris, 1 mM phenylmethanesulfonylfluoride, and a protease inhibitor cocktail (Sigma-Aldrich, St. Louis, MO). Immunoprecipitation was performed using magnetic protein G beads (Invitrogen) incubated with anti-Txnip (JY2) or mouse IgG antibodies (Sigma-Aldrich). Western analysis of bound proteins was performed with anti-VDUP1 antibody (Invitrogen) followed by HRP-conjugated anti-rabbit IgG VeriBlot antibody (Abcam).*

Comment: Figure 4, 5 and 6 demonstrated that Txnip deficiency exacerbated UPR. In figure 5 and 6, a negative regulatory role Txnip on UPR-Xbp-1s activation was revealed. But it's known

that Xbp-1s induces PDI expression as part of UPR. So it's good to know the expression level and activity of PDI under these conditions, to connect Txnip-PDI interaction and UPR more tightly.

Response: We performed experiments according to the reviewer's suggestions and included Pdi and Pdia6 expression results in Supporting Information Figure S3C-E. In addition we performed PDI activity assays under ER stress. We saw decreased PDI activity after addition of ER stress which can be explained by the multitude of intersecting signaling pathways that are activated by ER stress in the cell.

Supporting Information p. 5, para. 1: **C.** tunicamycin ($n = 4$) for 2 h. **D.** Relative transcript levels of Pdi and **E.** Pdia6 measured by qPCR normalized to 18S in serum-starved MEFs from WT and KO mice treated with tunicamycin ($n = 4$) for 2 h.

p. 10, para. 1: Since PDI and PDIA6 expression is induced via the UPR pathway (Hatahet & Ruddock, 2009), we also studied the effect of Txnip deficiency on Pdi and Pdia6 transcript levels in MEFs. Both at baseline and under ER stress, Pdi and Pdia6 transcript levels were significantly increased (Supporting Information Fig S3 C-E).

p. 21, para. 2: Primer sequences were from the MGH Primer Bank (Spandidos et al, 2010) or described previously (Oslowski & Urano, 2011): Xpb1s F: 5'-CTGAGTCCGAATCAGGTGCAG-3', Xpb1s R: 5'-GTCCATGGGAAGATGTTCTGG-3', Erdj3 F: 5'-GTACCTCATCGGGACTGTGAT-3', Erdj3 R: 5'-CAGAACCTCATAAGCAGCACCC3', Serp1 F: 5'-GCAACGTCGCTAAGACCTC-3', Serp1 R: 5'-CATGCCCATCCTGATACTTTGAA-3', Edem1 F: 5'-CTACCTGCGAAGAGGCCG-3', Edem1 R: 5'-CTACCTGCGAAGAGGCCG-3', Pdi F: 5'-GCCGCAAACTGAAGGCAG-3', Pdi R: 5'-GGTAGCCACGGACACCATAC-3', Pdia6 F: 5'-AGCTGCACCTTCTTTCTAGCA-3', Pdia6 R: 5'-CAGGCCGTCACCTCTGAATAAC-3'.

Supporting Information p. 7, para. 1: Supporting Information Figure 5. ER stress affects PDI activity in vitro. HEK293TN cells were transfected with the indicated plasmids in the presence or absence of tunicamycin (1 µg/ml x 2 h) and lysates were used to perform enzymatic activity assays. PDI activity was measured using a coupled insulin reduction assay (n = 3).

p. 15, para. 1: Subjecting cells to ER stress increases PDI expression levels (Hatahet & Ruddock, 2009); but given the complex nature of the UPR it is not surprising that overall PDI activity decreased after subjecting cells to ER stress using tunicamycin (Supporting Information Fig S5).

Comment: Figure 6 demonstrated that metabolic phenotype of Txnip deficiency may be attributable to its regulation of UPR signaling. ER stress has been linked to lipogenesis and hepatic lipid accumulation, and administration of chemical chaperones can alleviate atherosclerotic lesions (Hotamisligil GS 2010). So can chemical chaperones diminish Txnip KO mice liver Xbp-1s activation?

Response: We thank the reviewer for this suggestion and we performed experiments in liver-specific Txnip-KO mice with and without treatment with chemical chaperones tauroursodeoxycholic acid (TUDCA) or 4-phenylbutyric acid (PBA) which diminished Xbp1s activation.

p. 35, para. 2: F. Relative transcript levels of *Xbp1s* in liver samples of liver-specific *Txnip*-KO mice (*Cre*⁺) and littermate controls (*Cre*⁻) treated with vehicle, tauroursodeoxycholic acid (TUDCA) or 4-phenylbutyric acid (PBA) for 21 days (*n* = 3-4). * *p* < 0.05, ** *p* < 0.01 vs. *Cre*⁺ (vehicle only); *** *p* < 0.001 vs. *Cre*⁻ (vehicle only)

p. 12, para. 1: Next, we investigated whether these changes in ER stress signaling would also be reversible by treatment with TUDCA and PBA *in vivo*. We generated liver-specific *Txnip*-KO mice and confirmed increased *Xbp1* transcript levels compared to WT (Fig 7F). This increase was reversed by treatment with chemical chaperones (Fig 7F).

p. 22, para. 1: Total *Txnip*-KO mice and liver-specific *Txnip*-KO mice were described previously (Chutkow *et al*, 2008; Yoshioka *et al*, 2007). Mice were housed under controlled temperature and lighting with free access to water and food. Mice were maintained in accordance with the Institutional Animal Care and Use Committees of the Harvard School of Medicine. For *in vivo* studies, male 8-12-week-old mice received intraperitoneal injections of 250 mg/kg TUDCA or vehicle twice a day for 21 days; PBA was given in the drinking water at a concentration of 20 mM for 21 days.

p. 22, para. 2: All data are presented as mean ± SEM. Differences between groups were tested by unpaired two-tailed Student's t-test. Differences between multiple groups were tested by one-way ANOVA followed by post hoc analysis with the Tukey-Kramer test. A p-value less than 0.05 was considered statistically significant.

Comment: The discussion is not very clear on the relationship between the negative feedback of TXNIP on UPR and the phenotype of the TXNIP KO mouse.

Response: We revised the discussion according to the reviewer's suggestion.

p. 13, para. 3: Recent reports showed that Txnip itself is induced by ER stress, serving as a link between ER stress and inflammatory signaling (Lerner et al, 2012; Osowski et al, 2012). Activation of IRE1 α leads to reduction of the Txnip mRNA destabilizing microRNA-17 (Lerner et al, 2012). This results in increased levels of both Txnip mRNA as well as Txnip protein upon stimulation with different ER stress inducing agents. Txnip subsequently activates the NLRP3 inflammasome, which results in increased interleukin 1 β secretion (Zhou et al, 2010). These studies indicate that Txnip is an ER stress sensitive gene that is involved in linking UPR signaling with inflammatory activation. While increased ER stress levels induce Txnip expression, our results show that increased Txnip levels ultimately lead to decreased Xbp1s levels, the downstream signaling molecule of IRE1 α . Taken together these data reveal a negative feedback mechanism by which increased IRE1 α activation leads to induction of Txnip, which subsequently results in increased PDI activity and downregulation of IRE1 α signaling. Since Txnip is one of the

most strongly upregulated genes in diabetes (Parikh et al, 2007), this reveals Txnip as a key negative feedback regulatory mechanism of the UPR to decrease Xbp1s levels via direct regulation of PDI.

Being under posttranscriptional regulation by the IRE1 α pathway and affecting its downstream signaling molecule Xbp1s suggests that Txnip is involved in downstream signaling of this pathway as well. Several studies have shown that Xbp1s decreases gluconeogenesis and increases lipogenesis, adipogenesis, and insulin sensitivity (Lee et al, 2008; Ozcan et al, 2004; Sha et al, 2009; Zhou et al, 2011). Interestingly, Txnip deficiency in mice is associated with decreased gluconeogenesis, increased lipogenesis, adipogenesis, and insulin sensitivity—similar to the Xbp1s metabolic phenotype (Chutkow et al, 2010; Chutkow et al, 2008; Donnelly et al, 2004). The present study therefore provides evidence that some of the metabolic phenotypes of Txnip deficiency may be attributable to its regulation of UPR signaling.

Reviewer #3

In this paper the authors have studied protein disulfide isomerases (PDIs) and their interaction with the thioredoxin-interacting protein (Txnip). Txnip is known to be induced by glucose and bind to thioredoxin via a mixed disulfide formation involving Cys247 of Txnip and N-terminal active site Cys residue of Trx. The authors have shown that Txnip binds also to PDIs, which have two Trx domains with the sequence -CGHC-. They have convincingly shown that Txnip

indeed forms a covalent complex with both the intact PDI and the isolated Trx domains of several PDIs. Through mutational analysis they show that it is the N-terminal active site Cys residue, which is known to be the enzymatically active by attacking disulfide Cys residue, which is the target of Txnip.

Comment: On p 6 they then go on to show that Txnip increases the activity of PDI. The argument being that Txnip would shift PDI redox equilibrium to the reduced state thereby increasing its capacity they reduce other proteins. It is unclear how the authors mechanistically would explain this.

Response: We agree with the reviewer that the rather counterintuitive results of our PDI activity assays need to be explained in greater detail in our manuscript. We therefore added a new section to our Discussion explaining the rationale behind our hypothesis that Txnip increases PDI activity in our insulin reduction and turbidity assays.

p. 14, para. 3: It is somewhat surprising that Txnip, which has been characterized as an inhibitor of thioredoxin activity, increases activity of the thioredoxin fold-containing PDI. However, PDI primarily functions as an oxidase and has a higher standard reduction potential ($E_r^{0'} = -180 \text{ mV}$) than denatured proteins ($E_r^{0'} = -220 \text{ mV}$) while thioredoxin that functions as a reductase has a lower standard reduction potential ($E_r^{0'} = -270 \text{ mV}$) (Hatahet & Ruddock, 2009; Krause et al, 1991; Lundstrom & Holmgren, 1993). We therefore hypothesized that Txnip, being oxidized by PDI, shifts the PDI redox

equilibrium to the reduced state, thereby increasing its capacity to reduce other proteins, which was confirmed in our in vitro PDI activity assays.

Comment: Their results in Fig 3 A and 3 B is using cell extracts either over-expressing PDI or also Txnip with an artificial assay with high glutathione (8 mM) and insulin in the presence of a large excess of glutathione reductase and NADPH. Under these conditions, the extract containing PDI + Txnip, somewhat higher activity than the one with only PDI. The problem with this assay is that it is done with crude extract of cells, which also contain additional proteins like thioredoxin reductase and thioredoxin as well as glutaredoxins, which could interfere in the assay.

Response: We thank the reviewer for the insightful comments on an important limitation of our study. We agree with the reviewer that enzymatic assays should ideally be done using purified proteins. We performed numerous extensive experiments to purify Txnip using different approaches, including expression in *E. coli* or baculovirus-infected insect cells. However, we were only able to purify Txnip in its denatured form, which is not suitable for enzymatic assays. We added some of the data to the new Supporting Information Figures S6 and S7. Purification, and ideally crystallization, of Txnip protein is a major goal in the arrestin field, which has eluded a number of laboratories so far. Only crystallization of the N-terminal domain of human Txnip has been published so far (Polekhina et al, 2011). We have previously shown that results from enzymatic assays using crude cell extracts correlate well with in vitro assays using purified proteins (Holmgren & Bjornstedt, 1995; Patwari et al, 2006). Including mock transfected cells as a negative control allows for the most accurate assessment of Txnip function in enzymatic assays

at this point in time. We hope that the reviewer agrees with us that taken together we present a compelling case that Txnip increases PDI activity.

Supporting Information p. 8, para. 1: Supporting Information Figure 6. Txnip purification. *A.* Schematic representation of recombinant human Txnip (hTxnip) with an N-terminal 6xHistidine-tag (6xHis) in pTrcHis TOPO (Invitrogen). *B.* Coomassie staining and *C.* Western analysis of protein lysates from *E. coli* overexpressing hTxnip (pellet), from supernatant of *E. coli* cultures (supernatant), and purified Txnip in denaturing conditions. Recombinant hTxnip protein was purified in denatured condition;

however, solubility was limited in the supernatant fraction. **D.** Schematic representation of recombinant human Txnip with an N-terminal 6xHis-tag in fusion with *E. coli* Thioredoxin (*ecTrx*) in *pET-32a(+)* (Novagen). This system was used to enhance the production and solubility of *hTxnip*. **E.** Western analysis of protein levels of *ecTrx-6xHis-hTxnip* fusion protein in *E. coli* lysates after increasing durations of induction with Isopropyl β -D-1-thiogalactopyranoside (IPTG). **F.** Coomassie staining of *ec-Trx-6xHis-hTxnip* fusion protein after Ni-NTA-column purification and subsequent SDS-PAGE before and after thrombin cleavage with corresponding Western analyses using an anti-Txnip (JY2) and an anti-His-tag antibody. Although *hTxnip* protein was successfully purified in native condition with this method, there were multiple bands in Coomassie staining, indicating significant amounts of contamination by other proteins.

Supporting Information p. 11, para. 1: Supporting Information Figure 7. Txnip aggregation. A. Chromatogram of ecTrx-6xHis-hTxnip fusion protein after thrombin cleavage and gel filtration high performance liquid chromatography (HPLC). B. Coomassie staining of SDS-PAGE of fractions 5/6 (hTxnip) and 20 (ecTrx). Dynamic light scattering (DLS) demonstrated that the particle size of purified C. 6xHis-hTxnip and D. ecTrx-6xHis-hTxnip proteins was over 286 MDa and 181 MDa even with maximal usage of DTT. Thus, after purifying hTxnip protein by HPLC, hTxnip (fraction 5 and 6) formed protein aggregates.

Comment: In order to show that Txnip increases PDI activity, the authors should use a refolding assay of a protein like ribonuclease preferentially with pure protein and understand that conditions whereby the activity of ribonuclease is measured. In those conditions they would be able to show that Txnip indeed increases catalytic activity of PDI. As now, it is hard to understand how the binding of Txnip, which inhibits Trx activity, by binding the same way to PDI at the active site increase its catalytic activity.

Response: We agree with the reviewer that addition of another assay not involving insulin reduction was necessary to strengthen the evidence that Txnip increases PDI activity. However, since the use of crude cell extracts prevented us from performing the suggested ribonuclease isomerization assay, we performed a rhodanese chaperone assay, which has been used in multiple studies to measure PDI activity (Hatahet & Ruddock, 2009; Song & Wang, 1995; Uehara et al, 2006). This assay does not involve any disulfide exchange reactions; and Txnip

increased PDI activity in this assay as well. However, we still agree with the reviewer that even the results of our rhodanese chaperone assay highlight the limitations of our study in terms of using crude cell extracts instead of pure protein. Therefore, we added a new section in our Discussion to address this issue. We hope that the reviewer agrees with us that the revised version of our manuscript contains sufficient additional data to strengthen the evidence that Txnip increases PDI activity.

p. 34, para. 1: C. PDI activity was measured using a rhodanese aggregation assay (n = 3).

p. 8, para. 1: In addition to catalyzing disulfide exchange reactions, PDI also functions as a chaperone for proteins that do not contain any disulfide bonds, such as rhodanese (Hatahet & Ruddock, 2009). To investigate the effect of Txnip on PDI chaperone activity, we performed a rhodanese aggregation assay in which aggregation of denatured rhodanese is monitored by increase in absorbance at 320 nm. We hypothesized that PDI would decrease aggregation, but surprisingly PDI led to an increase in aggregation of

rhodanese, which was enhanced through addition of Txnip (Fig 4C). This is most likely due to the fact that we used protein extracts from cells overexpressing PDI and Txnip and will be discussed in more detail below.

p. 20, para. 2: *The assay was performed as described previously (Song & Wang, 1995). In this assay, aggregation of denatured rhodanese induced by guanidine hydrochloride is measured. 90 μ M rhodanese was denatured with 6 M guanidine hydrochloride for 30 min at room temperature and then diluted with cell lysates of HEK293TN cells transfected with the indicated plasmids in a buffer containing 30 mM Tris, 50 mM KCl, pH 7.2. Aggregation of rhodanese was measured by increase in absorbance at 320 nm.*

p. 15, para. 1: *However, we were surprised to find that in our rhodanese chaperone assay PDI increased rhodanese aggregation. Again, Txnip increased PDI activity in this assay as well, but we expected PDI to decrease rhodanese aggregation as had been described previously using purified proteins (Song & Wang, 1995; Uehara et al, 2006). These findings highlight a limitation of the present study, which is the use of crude cell extracts for PDI activity assays, which also contain additional proteins like thioredoxin reductase and thioredoxin as well as glutaredoxins, which could interfere in the assay. Ideally, purified proteins should be used in enzymatic assays. However, despite efforts by multiple laboratories including our own to purify Txnip protein (Supporting Information Fig S6 and Supporting Information Fig S7), only crystallization of the N-terminal domain of human Txnip has been published so far (Polekhina et al, 2011). We have previously shown that results from enzymatic assays using crude cell extracts correlate well with in*

vitro assays using purified proteins (Holmgren & Bjornstedt, 1995; Patwari et al, 2006). Including mock transfected cells as a negative control allows for the most accurate assessment of Txnip function in enzymatic assays at this point in time.

Comment: In Fig 3 B they use an insulin turbidity assay using a very low concentration of DTT of 10 μ M (should not this be 10 mM?). Is this indeed correct to get insulin precipitation? Again, this is an indirect assay of PDI activity using crude extracts but working in the wrong direction namely reducing a protein. Thus the authors, to convincingly show that PDI activity is increased, they should repeat the experiments with pure protein and using a refolding assay to demonstrate their point of increased analytic activity. The next step of course will be how to explain the binding of Txnip to PDI can lead to increased catalytic activity.

Response: We appreciate the attention to detail and level of expertise that the reviewer's comments display. We apologize for the mistake that we made in our Methods section regarding DTT concentration in our insulin turbidity assay which was a result of a typo in our assay protocol; we actually used 1 mM DTT according to previously published experiments (Holmgren, 1979). We corrected the Methods section accordingly.

p. 20, para. 1: HEK293TN cells were transfected with the indicated plasmids; cell lysates and recombinant PDI were added to a reaction mix with the final concentrations of 30 μ M insulin, 20 μ g recombinant PDI, and 1 mM DTT, in a buffer containing 84 mM sodium phosphate, 2.67 mM sodium EDTA, pH 7.0. Precipitation of insulin was measured by increase in absorbance at 650 nm.

Thank you for the submission of your manuscript to EMBO Molecular Medicine. We have now heard back from the three referees whom we asked to re-evaluate your manuscript.

As you can see from the enclosed reports, while referees 2 and 3 are now supportive of publication, referee 1 is still concerned about some important technicalities that must be resolved satisfactorily as to our opinion, the issues raised are central to the content of the paper as PDI is an ER-protein. As such, while we usually only allows one round of revision, due to the interesting nature of the data, we are ready to allow you to address the remaining issues. Needless to say, the manuscript will go back to this referee and acceptance will depend on the satisfaction of this last referee.

I look forward to seeing a revised form of your manuscript as soon as possible.

***** Reviewer's comments *****

Referee #1 (Remarks):

This reviewer thinks that the manuscript is not suitable for publication unless the authors can address the following concerns:

(Major points)

1. The author's response to the reviewer's comment 2: the author described that the HA epitope tag of PDI was added at the N-terminus of the signal sequence (p16, paragraph 2). This is quite problematic for the correct interpretation of the authors' results in this manuscript. If these constructs of PDIs would be correctly transported into the ER lumen as described by the authors, HA-signal sequence of PDI would be excised from the mature form of PDIs. Therefore these mature form-PDIs were not detected by anti-HA Ab. From this point, HA-tagged version of PDIs detected by pulldown experiments in Figs 1 and 2 might be located in the cytosol. The author should still confirm that the tagged versions of PDIs are indeed localized within the ER and that they are not in the cytosol
2. In the new Figure 3D, the authors introduced an N-glycosylation site into Txnip and examined whether Txnip entered the ER to interact with PDIs. This reviewer is not convinced with the conclusion of the authors from this panel that Txnip indeed entered the ER. It is not clear whether the digestion with PNGase F resulted in the appearance of a band that corresponds to the non-glycosylated form of this protein in panel D that should migrate faster than the N-glycosylated form: it looks as if one diffused band became one sharp band after the digestion. This appears to be due to the fact that, in contrast to panel C, the separation of bands is very poor with this panel. Thus, the authors should mark the positions of the N-glycosylated and non-glycosylated forms of Txnip in this panel after separating the related bands more clearly.
3. Fig.3B: The author should more clearly define the blotting band of PDIA6 in IP product. This reviewer did not [understand] whether this thin extended band was real positive signal or not.

(Minor point)

This reviewer is not sure whether the inclusion of Figure 4C is appropriate or not for reasons. Firstly, the authors concluded that the results of the assay yielded unexpected results due to the use of unpurified proteins. Although the authors speculated possible reasons for the unexpected results in the discussion section, this reviewer feels that more experimentation may be required for the valid interpretation of Figure 4C. Secondly, rhodanese aggregation assays using Txnip (in the absence of PDI) may be required, as a negative control, to conclude that Txnip increases the activity of PDI in promoting the aggregation of rhodanese.

Referee #2 (Remarks):

The authors have responded well to the previous comments

Referee #3 (Comments on Novelty/Model System):

After extensive revision a significant contribution.

Referee #3 (Remarks):

After extensive revision and discussion a significant contribution.

We appreciate the additional comments of the reviewer. We revised the manuscript according to the reviewer's suggestions and performed extensive additional experiments, which significantly improved the manuscript. Below, please find a point-by-point response to the issues brought up by the reviewer.

This reviewer thinks that the manuscript is not suitable for publication unless the authors can address the following concerns:

(Major points)

Comment: The author's response to the reviewer's comment 2: the author described that the HA epitope tag of PDI was added at the N-terminus of the signal sequence (p16, paragraph 2). This is quite problematic for the correct interpretation of the authors' results in this manuscript. If these constructs of PDIs would be correctly transported into the ER lumen as described by the authors, HA-signal sequence of PDI would be excised from the mature form of PDIs. Therefore these mature form-PDIs were not detected by anti-HA Ab. From this point, HA-tagged version of PDIs detected by pulldown experiments in Figs 1 and 2 might be located in the cytosol. The author should still confirm that the tagged versions of PDIs are indeed localized within the ER and that they are not in the cytosol.

Response: We thank the reviewer for raising this important issue. We would like to emphasize that full length PDIA6 pulldowns were always detected with an anti-PDIA6 antibody, not an HA-antibody. Also, since we performed pulldown experiments with non-PDI-transfected cells,

we could show that Txnip interacts with endogenous PDIA6, which does not contain an HA-tag (Fig 3B). According to the reviewer's suggestion we performed immunofluorescence analyses showing that HA-tagged PDI is still localized in the ER.

Supporting Information Figure 1

p. 7, para. 1: An important potential limitation of our pulldown assay is the use of PDI constructs with an N-terminal HA-tag which could interfere with intracellular trafficking. Therefore, we performed immunofluorescence analyses to confirm that HA-tagged PDI is still located in the ER (Supporting Information Fig S1A and B).

Supporting information p. 3, para. 1: Supporting Information Figure 1. HA-tagged PDI is localized in the ER. HEK293TN cells were transfected with HA-tagged PDI (+)

or an empty vector (-). Cells were subsequently fixed, permeabilized and stained for HA or ERp72 (ER). Scale bar = 25 μ m. A. Protein levels of HA-tagged PDI and ER marker ERp72 visualized by immunofluorescence under confocal microscopy with B. negative controls, probed with secondary antibodies only.

Comment: In the new Figure 3D, the authors introduced an N-glycosylation site into Txnip and examined whether Txnip entered the ER to interact with PDIs. This reviewer is not convinced with the conclusion of the authors from this panel that Txnip indeed entered the ER. It is not clear whether the digestion with PNGase F resulted in the appearance of a band that corresponds to the non-glycosylated form of this protein in panel D that should migrate faster than the N-glycosylated form: it looks as if one diffused band became one sharp band after the digestion. This appears to be due to the fact that, in contrast to panel C, the separation of bands is very poor with this panel. Thus, the authors should mark the positions of the N-glycosylated and non-glycosylated forms of Txnip in this panel after separating the related bands more clearly.

Response: We apologize for the poor quality of Figure 3D in the previous version of the manuscript. To try to address this issue biochemically, we have tried to purify functional Txnip but have been unsuccessful, using E. coli and eukaryotic expression. Since we show in Figure 3A that Txnip is glycosylated after addition of an N-glycosylation site as prescribed elsewhere (Kaup et al, 2011), we hope that the reviewer agrees that we have addressed the initial comment of the reviewer satisfactorily by performing the experiment that the reviewer had suggested.

Comment: Fig.3B: The author should more clearly define the blotting band of PDIA6 in IP

product. This reviewer did not [understand] whether this thin extended band was real positive signal or not.

Response: We apologize for the poor quality of Figure 3B in the previous version of the manuscript. Txnip is expressed at relatively low levels endogenously in this cell type and its molecular weight is similar to the heavy chain of IgG. Despite use of various preadsorbed and monoclonal secondary antibodies, co-immunoprecipitation is not a suitable method to yield data of sufficient quality for publication. We therefore performed extensive additional experiments to confirm that Txnip indeed interacts with endogenous PDIs. We chose an unbiased proteomics approach to identify potential protein-protein binding partners of Txnip. Among the top hits of this screen were several members of the PDI family. We included these data in the current version of the manuscript, as well as in a new supporting information spread sheet with the complete results of the proteomics screen.

Table 1. Mass spectrometry results for Txnip protein-protein interactions

Protein	MW (kDa)	Control (hits)	Txnip (hits)	
Txnip	44	-	2302	
Txn	12	-	388	Validation
Txn2	18	-	16	
Pdia4	73	-	72	Protein disulfide isomerases
Pdia6	48	-	25	
Pdia15	44	-	3	

Supporting Information Figure 2

p. 7, para. 1: To confirm these findings we chose an unbiased proteomics approach to identify protein-protein interaction partners for Txnip (Gao et al, 2009). Affinity chromatography with subsequent SDS-PAGE and mass spectrometry analyses confirmed that Txnip interacts with several endogenously expressed PDIs (Table 1 and Supporting Information Fig S2).

p. 18, para. 3:

Proteomics screen for Txnip protein-protein interactions

Txnip was subcloned into pCDH-CMV-MCS-EF1-Puro (System Biosciences) with an N-terminal tandem Strep/FLAG epitope tag

(mDYKDDDDKgsaasWSHPQFEKgggsgggsgggsWSHPQFEK) (Gloeckner et al, 2009).

HEK293F cells were stably transduced with Txnip or an empty vector control and clonally selected; cell culture was scaled up in suspension cell culture (Integra). Affinity chromatography was performed using Streptactin resin columns (IBA) according to the manufacturer's instructions. Eluates were subjected to SDS-PAGE and mass spectrometry analysis as described previously (Gao et al, 2009; Yoshioka et al, 2012).

Supporting information p. 4, para. 1: Supporting Information Figure 2. Txnip affinity chromatography and SDS-PAGE. Affinity chromatography was performed to purify Txnip, and potential protein-protein interaction partners, from HEK293F cells stably transduced with SF:Txnip or an empty control vector. Eluates were subjected to SDS-PAGE, Coomassie-stained bands were excised and analyzed by mass spectrometry.

(Minor point)

Comment: This reviewer is not sure whether the inclusion of Figure 4C is appropriate or not for reasons. Firstly, the authors concluded that the results of the assay yielded unexpected results due to the use of unpurified proteins. Although the authors speculated possible reasons for the unexpected results in the discussion section, this reviewer feels that more experimentation may be required for the valid interpretation of Figure 4C. Secondly, rhodanese aggregation assays using Txnip (in the absence of PDI) may be required, as a negative control, to conclude that Txnip increases the activity of PDI in promoting the aggregation of rhodanese.

Response: In accordance with the suggestion of the reviewer, we did not include Figure 4C in the final version of the manuscript.

Thank you for the submission of your revised manuscript to EMBO Molecular Medicine. We have now received the enclosed report from the referee who was asked to re-assess it. As you will see, this reviewer remains concerned about the data. As such, we extensively discussed the issue within the team, including our Chief Editor and I am pleased to inform you that we will be able to accept your manuscript pending final editorial amendments:

Please submit your revised manuscript within two weeks. I look forward to seeing a revised form of your manuscript as soon as possible.

***** Reviewer's comments *****

Referee #1 (Comments on Novelty/Model System):

The reason why this manuscript is not suitable for publication is described in remarks to the authors. This paper has two big problems. One is the mistake of the construction of HA-tagged version of PDI families, which means the HA-tag was fused to the N-terminus of signal sequence. The other one is that there are little evident data and poor explanation about how Txnip could be transported into ER lumen. Unless the authors would present more clear and absolute data, I cannot agree with the publication of this manuscript.

Referee #1 (Remarks):

One of the important claims in this paper is that Txnip is localized in the inside of the ER to influence the activity of PDI family members. Since this protein does not seem to have a conventional ER targeting signal such as N-terminal signal sequence as I have commented before, whether this protein is indeed localized in the ER or not has to be examined carefully. I think the results are not so convincing to conclude that this protein is localized in the inside of the ER as I will explain below. Thus, I feel that this paper is not suitable for publication in EMBO Molecular Medicine.

1. Figure 3A:

To examine whether Txnip is located within the ER, the authors introduced N-glycosylation sequence into Txnip. Upon expressing the construct, the authors observed two bands for Txnip. The authors claim that the upper band corresponds to the N-glycosylated form of Txnip and the lower band the unglycosylated form. If this interpretation is correct, the upper band should be converted into the lower band upon treatment of the lysate with N-glycosidase F. However, the authors have not obtained clear data that support this conclusion.

2. Supporting Information Figure 1:

Most of the pull down assays were performed using PDIs having an HA tag at the N-terminus of their signal peptide. Because of the problem I will explain below in the item #3, I have a serious concern about the localization of the HA-tagged PDIs.

To show that the HA-tagged PDIs are localized to the ER, the authors tested the location of the protein by immuno-fluorescence microscopy. The signal obtained with anti-HA antibody overlapped with the signal obtained with antibody against ERp72 (a ER marker). This result suggests the HA-tagged PDI is localized to the ER. However, since there still remains a possibility that HA-tagged PDI is located on the surface of the ER, I think this result does not verify that HA-tagged PDIs are really exported into the inside of the ER.

3. Figure 1, panels B-D; Figure 2, panels A-C; Figure 3B

As already described above, most of the pull down assays were performed using PDIs having an HA tag at the N-terminus of their signal peptide. The signal peptides are usually cleaved when the

proteins are exported into the ER. Thus, the HA-tagged PDIs are expected to lose the HA-tag when they are exported into the ER. The authors detected PDIs using HA antibody. This means that the detected PDIs are precursor proteins still having a signal sequence. This raises a concern that these PDIs may not have entered the ER and the detected complexes between HA-tagged PDIs and Txnip may have been formed in the cytosol in contrast to the claim of the authors that the complexes were formed in the ER.

3rd Revision - authors' response

17 April 2014

We thank the editors for the additional comments. We revised the manuscript according to the editors' suggestions, which significantly improved the manuscript.